# *CellFlux*: Simulating Cellular Morphology Changes via Flow Matching

**Yuhui Zhang** [1] [*]  **Yuchang Su** [2] [*]  **Chenyu Wang** [3]  **Tianhong Li** [3]  **Zoe Wefers** [1]  **Jeffrey Nirschl** [1]  **James Burgess** [1]
**Daisy Ding** [1]  **Alejandro Lozano** [1]  **Emma Lundberg** [1]  **Serena Yeung-Levy** [1]

## Abstract

Building a virtual cell capable of accurately simulating cellular behaviors in silico has long been a dream in computational biology. We introduce *CellFlux*, an image-generative model that simulates cellular morphology changes induced by chemical and genetic perturbations using flow matching. Unlike prior methods, *CellFlux* models distribution-wise transformations from unperturbed to perturbed cell states, effectively distinguishing actual perturbation effects from experimental artifacts such as batch effects—a major challenge in biological data. Evaluated on chemical (BBBC021), genetic (RxRx1), and combined perturbation (JUMP) datasets, *CellFlux* generates biologically meaningful cell images that faithfully capture perturbation-specific morphological changes, achieving a 35% improvement in FID scores and a 12% increase in mode-of-action prediction accuracy over existing methods. Additionally, *CellFlux* enables continuous interpolation between cellular states, providing a potential tool for studying perturbation dynamics. These capabilities mark a significant step toward realizing virtual cell modeling for biomedical research. Project page: https://yuhui-zh15.github.io/CellFlux/.

## 1. Introduction

Building a virtual cell that simulates cellular behaviors in silico has been a longstanding dream in computational biology (Slepchenko et al., 2003; Johnson et al., 2023; Bunne et al., 2024). Such a system would revolutionize drug discovery by rapidly predicting how cells respond to new compounds or genetic modifications, significantly reducing the cost and time of biomedical research by prioritizing the experiments most likely to succeed based on the virtual cell simulation (Carpenter, 2007). Moreover, this could unlock personalized therapeutic development by building digital twins of cells from patients to simulate patient-specific responses (Katsoulakis et al., 2024).

Two recent advances have made creating a generative virtual cell model possible. On the computational side, generative models now excel at modeling and sampling from complex data distributions, demonstrating remarkable success in synthesizing texts, images, videos, and biological sequences (OpenAI, 2024; Esser et al., 2024; Kondratyuk et al., 2024; Hayes et al., 2025). Concurrently, on the biotechnology side, automated high-content screening has generated massive imaging datasets — reaching terabytes or petabytes — that capture how cells respond to hundreds of thousands of chemical compounds and genetic modifications (Chandrasekaran et al., 2023; Fay et al., 2023).

In this work, we introduce *CellFlux*, an image-generative model that simulates how cellular morphology changes in response to chemical or genetic perturbations (Figure 1a). *CellFlux*'s key innovation is formulating cellular morphology prediction as a distribution-to-distribution learning problem, and leveraging flow matching (Lipman et al., 2023), a state-of-the-art generative modeling technique designed for distribution-wise transformation, to solve this problem.

Specifically, cell morphology data are collected through high-content microscopy screening, where images of control and perturbed cells are captured from experimental wells across different batches (Figure 1b). Control wells, which receive no drug treatment or genetic modifications, play a crucial role in providing prior information and serving as a reference to distinguish true perturbation effects from other sources of variation. They help calibrate non-perturbation factors, such as batch effects—systematic biases unrelated to perturbations, including variations in color or intensity, akin to distribution shifts in machine learning. Properly incorporating control wells is essential for capturing actual perturbation effects rather than artifacts, yet many existing methods overlook this aspect (Yang et al., 2021; Navidi et al., 2025; Cook et al., 2024). To address this, we frame cellular morphology prediction as a distribution-to-distribution

---
[*]Equal contribution  [1]Stanford University  [2]Tsinghua University  [3]MIT. Correspondence to: Yuhui Zhang <yuhuiz@stanford.edu>, Serena Yeung-Levy <syyeung@stanford.edu>.

*Proceedings of the 42nd International Conference on Machine Learning*, Vancouver, Canada. PMLR 267, 2025. Copyright 2025 by the author(s).

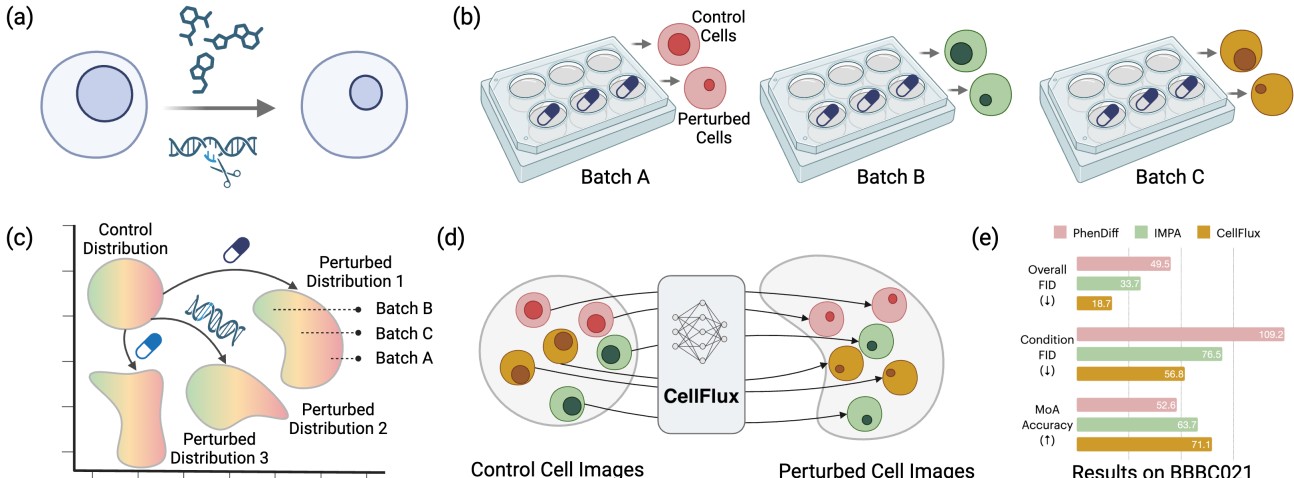

Figure 1. **Overview of *CellFlux*.** *(a) Objective. CellFlux* aims to predict changes in cell morphology induced by chemical or gene perturbations *in silico*. In this example, the perturbation effect reduces the nuclear size. *(b) Data.* The dataset includes images from high-content screening experiments, where chemical or genetic perturbations are applied to target wells, alongside control wells without perturbations. Control wells provide prior information to contrast with target images, enabling the identification of true perturbation effects (e.g., reduced nucleus size) while calibrating non-perturbation artifacts such as batch effects—systematic biases unrelated to the perturbation (e.g., variations in color intensity). *(c) Problem formulation.* We formulate the task as a distribution-to-distribution problem (many-to-many mapping), where the source distribution consists of control images, and the target distribution contains perturbed images within the same batch. *(d) Flow matching. CellFlux* employs flow matching, a state-of-the-art generative approach for distribution-to-distribution problems. It learns a neural network to approximate a velocity field, continuously transforming the source distribution into the target by solving an ordinary differential equation (ODE). *(e) Results. CellFlux* significantly outperforms baselines in image generation quality, achieving lower Fréchet Inception Distance (FID) and higher classification accuracy for mode-of-action (MoA) predictions.

mapping problem (Figure 1c), where the source distribution consists of control cell images, and the target distribution comprises perturbed cell images from the same batch.

To address this distribution-to-distribution problem, *CellFlux* employs flow matching, a state-of-the-art generative modeling approach designed for distribution-wise transformations (Figure 1d). The framework continuously transforms the source distribution into the target using an ordinary differential equation (ODE) by learning a neural network to approximate a velocity field. This direct and native distribution transformation enabled by flow matching is intuitively more effective than previous methods, which rely on adding extra components to GANs, incorporating the source as a condition, or mapping between distributions and noise using diffusion models (Palma et al., 2025; Hung et al., 2024; Bourou et al., 2024).

We demonstrate the effectiveness of *CellFlux* on three datasets: BBBC021 (chemical perturbations) (Caie et al., 2010), RxRx1 (genetic modifications via CRISPR or ORF) (Sypetkowski et al., 2023), and JUMP (combined chemical and genetic perturbations) (Chandrasekaran et al., 2023). *CellFlux* generates high-fidelity images of cellular changes in response to perturbations across all datasets, improving FID scores by 35% over previous approaches. The generated images capture meaningful biological patterns, demonstrated by a 12% improvement in predicting mode-

of-action compared to existing methods (Figure 1e). Importantly, *CellFlux* maintains consistent performance across diverse experimental conditions and generalizes to held-out perturbations never seen during training, showing its broad applicability.

Moreover, *CellFlux* introduces two key capabilities with significant potential for biological research (Figure 4). First, it effectively corrects batch effects by conditioning on control cells from different batches. By comparing control images with generated images, it can disentangle true perturbation-induced morphological changes from experimental batch artifacts. Second, *CellFlux* enables bidirectional interpolation between cellular states due to the continuous and reversible nature of the velocity field in flow matching. This interpolation provides a means to explore intermediate cellular morphologies and potentially gain deeper insights into dynamic perturbation responses.

In summary, by formulating cellular morphology prediction as a distribution-to-distribution problem and using flow matching as a solution, *CellFlux* enables accurate prediction of perturbation responses (Figure 1). *CellFlux* not only achieves state-of-the-art performance but unlocks new capabilities such as handling batch effects or visualizing cellular state transitions, significantly advancing the field towards a virtual cell for drug discovery and personalized therapy.

## 2. Problem Formulation

In this section, we introduce the objective, data, and mathematical formulation of cellular morphology prediction.

### 2.1. Objective

Let $\mathcal{X}$ denote the cell image space and $\mathcal{C}$ the perturbation space. Let $p_0$ represent the original cell distribution and $p_1$ represent the distribution of cells after a perturbation $c \in \mathcal{C}$. Cellular morphology prediction aims to learn a generative model $p_\theta : \mathcal{X} \times \mathcal{C} \to \mathcal{P}(\mathcal{X})$, which, given an unperturbed cell image $x_0 \sim p_0$ and a perturbation $c \in \mathcal{C}$, predicts the resulting conditional distribution $p(x_1|x_0, c)$. From this distribution, new images can be sampled to simulate the effects of the perturbation, such that $x_1 \sim p_1$ (Figure 1a).

The input space consists of multi-channel microscopy images, where $\mathcal{X} \subset \mathbb{R}^{H \times W \times C}$. Here, $H$ and $W$ represent the image height and width, while $C$ denotes the number of channels, each highlighting different cellular components through specific fluorescent markers (analogous to RGB channels in natural images, but capturing biological structures like mitochondria, nuclei, and cellular membranes).

The perturbation space $\mathcal{C}$ includes two types of biological interventions: chemical (drugs) and genetic (gene modifications). Chemical perturbations involve compounds that target specific cellular processes — for example, affecting DNA replication or protein synthesis. Genetic perturbations can turn off gene expression (CRISPR) or upregulate gene expression (ORF).

This generative model enables in silico simulation of cellular responses, which traditionally require time-intensive and costly wet-lab experiments. Such computational modeling could revolutionize drug discovery by enabling rapid virtual drug screening and advance personalized medicine through digital cell twins for treatment optimization.

### 2.2. Data

Cell morphology data are collected through high-content microscopy screening (Figure 1b) (Perlman et al., 2004). In this process, biological samples are prepared in multi-well plates containing hundreds of independent experimental units (wells). Selected wells receive interventions — either chemical compounds or genetic modifications — while control wells remain unperturbed. After a designated period, cells are fixed using chemical fixatives and stained with fluorescent dyes to highlight key structures like the nucleus, cytoskeleton, and mitochondria. An automated microscope then captures multiple images per well. This process is called cell painting. Modern automated high-content screening systems have enabled large-scale data collection, resulting in datasets of terabyte to petabyte im-

ages from thousands of perturbation conditions (Fay et al., 2023; Chandrasekaran et al., 2023).

However, the cell painting process has limitations: cell painting requires cell fixation, which is destructive, making it impossible to observe the same cells dynamically during a perturbation. This creates a fundamental constraint: we cannot obtain paired samples $\{(x_0, x_1)\}$ showing the exact same cell without and with treatment. Instead, we must work with unpaired data $(\{x_0\}, \{x_1\})$, where $\{x_0\}$ represents control images and $\{x_1\}$ represents treated images, to learn the conditional distribution $p(x_1|x_0, c)$.

One solution is to leverage the distribution transformation from control cells to perturbed cells within the same batch to learn conditional generation. Control cells serve as a crucial reference by providing prior information to separate true perturbation effects from confounding factors such as batch effects. Variations in experimental conditions across different runs (batches) introduce systematic biases unrelated to the perturbation itself. For instance, images from one batch may consistently differ in pixel intensities from those in another. Therefore, meaningful comparisons require analyzing treated and control samples from the same batch. As shown in Figure 1b, this approach helps distinguish true biological responses, like changes in nuclear size, from batch-specific artifacts, like changes in color.

### 2.3. Mathematical Formulation

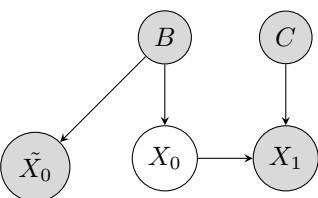

Let us formalize our learning problem in light of the experimental constraints described before. Our objective is to learn a conditional distribution $p(x_1|x_0, c)$ that models the cellular response to perturbation. However, due to the destructive nature of imaging, we cannot observe paired samples $\{(x_0, x_1)\}$. We propose a probabilistic graphical model to address this challenge.

In our graph, random variable $B$ denotes the experimental batch, $C$ denotes the perturbation condition, $X_0$ represents the unobservable basal cell state, $\tilde{X}_0$ represents control cells from the same batch, and $X_1$ denotes the perturbed cell state. From our experimental setup, we have access to the control distribution $p(\tilde{x}_0|b)$ from unperturbed cells and the perturbed distribution $p(x_1|c, b)$ from treated cells.

We propose to leverage the distributional transition from $p(\tilde{x}_0|b)$ to $p(x_1|c, b)$ to learn the individual-level trajectory $p(x_1|x_0, c)$, as shown in Figure 1c. There are two key reasons. First, there exists a natural connection be-

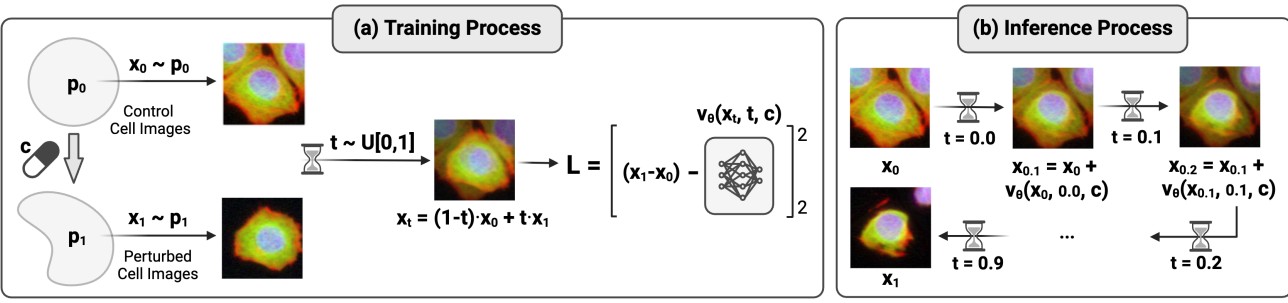

Figure 2. **CellFlux** algorithm. *(a) Training.* The neural network $v_\theta$ learns a velocity field by fitting trajectories between control cell images ($x_0 \sim p_0$) and perturbed cell images ($x_1 \sim p_1$). At each training step, intermediate states $x_t$ are sampled along the linear interpolation between $x_0$ and $x_1$, with $t \sim U[0, 1]$. The network minimizes the loss $L$, which measures the difference between the predicted velocity $v_\theta(x_t, t, c)$ and the true velocity ($x_1 - x_0$). *(b) Inference.* The trained velocity field $v_\theta$ guides the transformation of a control cell state $x_0$ into a perturbed cell state $x_1$. This is achieved by solving an ordinary differential equation iteratively, using numerical integration steps over time $t$ (e.g., $t = 0.0, 0.1, 0.2, \ldots, 0.9, 1.0$). Each step updates the cell state using the learned velocity field.

tween $p(x_1|c, b)$ and $p(x_0|b)$ through the marginalization $p(x_1|c, b) = \int p(x_1|x_0, c)p(x_0|b)dx_0$. Second, while $p(x_0|b)$ is not directly tractable, we can approximate it using $p(\tilde{x}_0|b)$ since both the ground-truth $X_0$ distribution and control distribution $\tilde{X}_0$ follow the same batch-conditional distribution: $x_0 \sim p(\cdot|b)$ and $\tilde{x}_0 \sim p(\cdot|b)$.

Our approach of learning $p(x_1|\tilde{x}_0, c)$ by conditioning on same-batch control images improves upon existing methods that ignore control cells and learn only $p(x_1|c)$. Intuitively, conditioning on $\tilde{x}_0$ allows the model to initiate the transition from a distribution more closely aligned with the underlying $x_0$, leading to a better approximation of the true distribution $p(x_1|x_0, c)$. We formalize this intuition in the following proposition, with proof provided in Appendix A:

**Proposition 1.** *Given random variables $B$, $C$, $X_0$, $\tilde{X}_0$, and $X_1$ following the graphical model above with joint distribution $p(b, c, x_0, \tilde{x}_0, x_1)$, the distribution $p(x_1|x_0, c)$ can be better approximated by the conditional distribution $p(x_1|\tilde{x}_0, c)$ than $p(x_1|c)$ in expectation. Formally,*

$$\mathbb{E}_{p(x_0, \tilde{x}_0, c)} [D_{KL}(p(x_1|x_0, c)||p(x_1|\tilde{x}_0, c))]$$
$$\leq \mathbb{E}_{p(x_0, c)} [D_{KL}(p(x_1|x_0, c)||p(x_1|c))]$$

## 3. Method

As detailed in §2, we predict cell morphological changes by transforming distributions between control and perturbed cells under specific conditions within the same batch. In this section, we introduce *CellFlux*, which leverages flow matching, a principled framework for learning continuous transformations between probability distributions. We adapt flow matching with condition, noise augmentation, and classifier-free guidance to better address our problem setting.

### 3.1. Preliminaries

Flow matching (Lipman et al., 2023; 2024) provides a framework to learn transformations between probability distribu-

tions by constructing smooth paths between paired samples (Figure 1d). It models how a source distribution continuously deforms into a target distribution through time, similar to morphing one shape into another.

More formally, consider probability distributions $p_0$ and $p_1$ defined on a metric space $(\mathcal{X}, d)$. Given pairs of samples from these distributions, flow matching learns a time-dependent velocity field using a neural network $v_\theta : \mathcal{X} \times [0, 1] \to \mathcal{X}$ that describes the instantaneous direction and magnitude of change at each point. The transformation process follows an ordinary differential equation:

$$dx_t = v_\theta(x_t, t)dt, \quad x_0 \sim p_0, \quad x_1 \sim p_1, \quad t \in [0, 1]$$

During training, we construct a probability path that connects samples from the source ($p_0$) and target ($p_1$) distributions (Figure 2a). We employ the rectified flow formulation, which yields a simple straight-line path (Liu et al., 2023):

$$x_t = (1 - t)x_0 + tx_1, \quad t \sim \mathcal{U}[0, 1]$$

This linear path has a constant velocity field $v(x_t, t) = dx_t/dt = x_1 - x_0$, which represents the optimal transport direction at each point. The neural network $v_\theta$ is trained to approximate this optimal velocity field by minimizing:

$$\mathcal{L}(\theta) = \mathbb{E}_{x_0 \sim p_0, x_1 \sim p_1, t \sim \mathcal{U}[0, 1]} \|v_\theta(x_t, t) - v(x_t, t)\|_2^2$$

At inference time, given a sample $x_0 \sim p_0$, we generate $x_1$ by solving the ODE (Figure 2b), whose solution is:

$$x_1 = x_0 + \int_0^1 v_\theta(x_t, t)dt$$

We employ numerical integrators like Euler method or more advanced methods such as Runge-Kutta to solve the ODE.

**(a) Main Results**

| Method | BBBC021 (Chemical) | | | | RxRx1 (Genetic) | | | | JUMP (Combined) | | | |
|---|---|---|---|---|---|---|---|---|---|---|---|---|
| | $\text{FID}_o$ | $\text{FID}_c$ | $\text{KID}_o$ | $\text{KID}_c$ | $\text{FID}_o$ | $\text{FID}_c$ | $\text{KID}_o$ | $\text{KID}_c$ | $\text{FID}_o$ | $\text{FID}_c$ | $\text{KID}_o$ | $\text{KID}_c$ |
| PhenDiff (MICCAI'24) | 49.5 | 109.2 | 3.10 | 3.18 | 65.9 | 174.4 | 5.19 | 5.29 | 49.3 | 127.3 | 5.09 | 5.17 |
| IMPA (Nature Comm'25) | 33.7 | 76.5 | 2.60 | 2.70 | 41.6 | 164.8 | 2.91 | 2.94 | 14.6 | 99.9 | 1.08 | 1.06 |
| CellFlux | **18.7** | **56.8** | **1.62** | **1.59** | **33.0** | **163.5** | **2.38** | **2.40** | **9.0** | **84.4** | **0.63** | **0.65** |

**(b) Per Perturbation Results**

| Method | Chemical Perturbations | | | | | | Genetic Perturbations | | |
|---|---|---|---|---|---|---|---|---|---|
| | Alsterpaullone | AZ138 | Bryostatin | Colchicine | Mitomycin C | PP-2 | ACSS1 | CRISP3 | RASD1 |
| PhenDiff (MICCAI'24) | 106.6 | 120.0 | 106.9 | 111.2 | 110.0 | 121.7 | 157.5 | 144.6 | 180.4 |
| IMPA (Nature Comm'25) | 69.6 | 59.9 | 104.3 | 84.4 | 57.0 | 77.3 | 152.6 | 142.7 | 147.1 |
| CellFlux | **41.6** | **44.4** | **47.0** | **72.3** | **42.3** | **64.3** | **140.9** | **125.1** | **140.1** |

*Table 1.* **Evaluation of *CellFlux*.** *(a) Main results. CellFlux* outperforms GAN- and diffusion-based baselines, achieving state-of-the-art performance in cellular morphology prediction across three chemical, genetic, and combined perturbations datasets. Metrics measure the distance between generated and ground-truth samples, with lower values indicating better performance. $\text{FID}_o$ (overall FID) evaluates all images, while $\text{FID}_c$ (conditional FID) averages results per perturbation $c$. KID values are scaled by 100 for visualization. *(b) Per perturbation results.* For six representative chemical perturbations and three genetic perturbations, *CellFlux* generates significantly more accurate images that better capture the perturbation effects than other methods, as measured by the FID score.

### 3.2. Conditional Flow Matching

To model perturbation conditions, we extend flow matching by conditioning on perturbations $c \in \mathcal{C}$. While the source distribution $p_0$ represents unperturbed cell images, the target distribution now becomes condition-dependent, denoted as $p_1(x|c)$. Our goal is to learn a conditional velocity field $v_\theta : \mathcal{X} \times [0,1] \times \mathcal{C} \to \mathcal{X}$ that captures perturbation-specific transformations (Esser et al., 2024):

$$dx_t = v_\theta(x_t, t, c)dt, \quad x_0 \sim p_0, \quad x_1 \sim p_1(\cdot|c)$$

### 3.3. Classifier-Free Guidance

We incorporate classifier-free guidance (Ho & Salimans, 2022; Zheng et al., 2023) to improve generation fidelity. During training, we randomly mask conditions with probability $p_c$, replacing $c$ with a null token $\emptyset$. At inference time, we interpolate between conditional and unconditional predictions:

$$v_\theta^{\text{CFG}}(x_t, t, c) = \alpha \cdot v_\theta(x_t, t, c) + (1-\alpha) \cdot v_\theta(x_t, t, \emptyset)$$

where $\alpha > 1$ controls guidance strength.

### 3.4. Noise Augmentation

Since $p_0$ and $p_1$ are both empirical distributions from datasets with limited observations, direct mapping between them may lead to bad generalization. Therefore, we propose augmenting the samples to make the learned velocity field smoother. This is done by adding random Gaussian noise to $x_0 \sim p_0$ with a probability $p_e$. Formally:

$$\tilde{x}_0 = \begin{cases} x_0 + \epsilon, & \text{with probability } p_e \\ x_0, & \text{with probability } 1 - p_e \end{cases}$$

where $\epsilon \sim \mathcal{N}(0, \sigma^2 I)$. This noise augmentation helps prevent overfitting to discrete samples and encourages the model to learn a continuous velocity field in the ambient space. The noise scale $\sigma$ and probability $p_e$ are hyperparameters that control the smoothness of the learned field.

### 3.5. Neural Network Architecture

The velocity field $v_\theta$ is realized through a U-Net architecture (Ronneberger et al., 2015), as we directly model the distribution in image pixel space $\mathcal{X} \subset \mathbb{R}^{H \times W \times C}$, where U-Net captures both local and global features through its multi-scale structure. Time $t$ is encoded using Fourier features, and condition $c \in \mathcal{C}$ is embedded through a learnable network $E : \mathcal{C} \to \mathbb{R}^d$. These embeddings are added to form the condition signal, which is then injected into the U-Net blocks to guide the generation process (Esser et al., 2024).

The entire *CellFlux* algorithm is summarized in §B.

## 4. Results

In this section, we present detailed results demonstrating *CellFlux*'s state-of-the-art performance in cellular morphology prediction under perturbations, outperforming existing methods across multiple datasets and evaluation metrics.

### 4.1. Datasets

Our experiments were conducted using three cell imaging perturbation datasets: BBBC021 (chemical perturbation) (Caie et al., 2010), RxRx1 (genetic perturbation) (Sypetkowski et al., 2023), and the JUMP dataset (combined perturbation) (Chandrasekaran et al., 2023). We followed the preprocessing protocol from IMPA (Palma et al., 2025), which involves correcting illumination, crop-

**(a) Mode of Action Classification**

| Method | MoA Accuracy | MoA Macro-F1 | MoA Weighted-F1 |
|---|---|---|---|
| Groundtruth Perturbed Image | 72.4 | 69.7 | 72.1 |
| PhenDiff | 52.6 | 33.6 | 52.1 |
| IMPA | 63.7 | 40.2 | 64.8 |
| CellFlux | **71.1** | **49.0** | **70.7** |

**(b) Out-of-Distribution Generalization**

| Method | $FID_o$ | $FID_c$ | $KID_o$ | $KID_c$ | MoA Accuracy | MoA Macro-F1 | MoA Weighted-F1 |
|---|---|---|---|---|---|---|---|
| Groundtruth Perturbed Image | 0.0 | 0.0 | 0.0 | 0.0 | 88.0 | 85.0 | 88.0 |
| PhenDiff | 67.7 | 151.6 | 3.45 | 3.71 | 9.6 | 9.3 | 7.4 |
| IMPA | 44.5 | 136.9 | 3.07 | 3.24 | 16.0 | 10.0 | 13.1 |
| CellFlux | **42.0** | **98.0** | **1.31** | **1.23** | **43.2** | **36.6** | **42.8** |

**(c) Batch Effect Study**

| Method | $FID_o$ | $FID_c$ | $KID_o$ | $KID_c$ | MoA Accuracy | MoA Macro-F1 | MoA Weighted-F1 |
|---|---|---|---|---|---|---|---|
| CellFlux w/ Other Batch Init | 19.9 | 66.3 | 1.70 | 1.69 | 48.2 | 32.9 | 48.4 |
| CellFlux | **18.7** | **56.8** | **1.62** | **1.59** | **71.2** | **49.0** | **70.7** |

**(d) Ablation Study**

| Method | $FID_o$ | $FID_c$ | $KID_o$ | $KID_c$ |
|---|---|---|---|---|
| CellFlux w/o Condition | 45.0 | 113.0 | 2.37 | 2.37 |
| CellFlux w/o CFG | 32.6 | 92.4 | **1.23** | 1.35 |
| CellFlux w/o Noise | 31.9 | 91.4 | 1.24 | **1.26** |
| CellFlux | **18.7** | **56.8** | 1.62 | 1.59 |

*Table 2.* **More evaluation and ablation of *CellFlux*.** *(a) MoA classification.* On the BBBC021 dataset, we train a classifier to predict the drug's mode of action (MoA) from cell morphology images and evaluate the accuracy/F1 of generated images. *CellFlux* achieves significantly higher accuracy/F1 than other methods, closely aligning with ground-truth images and effectively reflecting the biological effects of perturbations. *(b) Out-of-distribution generalization.* *CellFlux* maintains strong performance when generating cell morphology images for novel chemical compounds not seen during training on BBBC021. *(c) Batch effect study.* *CellFlux* shows improved performance when using control images from the same batch as initialization, highlighting the critical role of control images in calibrating batch effects. *(d) Ablation study.* Removing key components degrades *CellFlux*'s performance, emphasizing their importance.

ping images centered on nuclei to a resolution of 96×96, and filtering out low-quality images. The resulting datasets include 98K, 171K, and 424K images with 3, 6, and 5 channels, respectively, from 26, 1,042, and 747 perturbation types. Examples of these images are provided in Figure 3. Details of datasets are provided in §E.

### 4.2. Experimental Setup

**Evaluation metrics.** We evaluate methods using two types of metrics: (1) FID and KID (lower the better), which measure image distribution similarity via Fréchet and kernel-based distances, computed on 5K generated images for BBBC021 and 100 randomly selected perturbation classes for RxRx1 and JUMP; we report both overall scores across all samples and conditional scores per perturbation class. (2) Mode of Action (MoA) classification accuracy and F1 score (higher the better), which assesses biological fidelity by using a trained classifier to predict a drug's effect from perturbed images and comparing it to its known MoA from the literature.

**Baselines.** We compare our approach against two baselines, PhenDiff (Bourou et al., 2024) and IMPA (Palma et al., 2025), the only two baselines that incorporate control images into their model design — a crucial setup for distinguishing true perturbation effects from artifacts such as batch effects. PhenDiff uses diffusion models to first map control images to noise and then transform the noise into target images. In contrast, IMPA employs GANs with an AdaIN layer to transfer the style of control images to target images, specifically designed for paired image-to-image mappings. Our method uses flow matching, which is tailored for distribution-to-distribution mapping, providing a more suitable solution for our problem. We reproduce these baselines with official codes.

**Training details.** *CellFlux* employs a UNet-based velocity field with a four-stage design. Perturbations are encoded following IMPA (Palma et al., 2025). Training is conducted for 100 epochs on 4 A100 GPUs. Details are in §C.

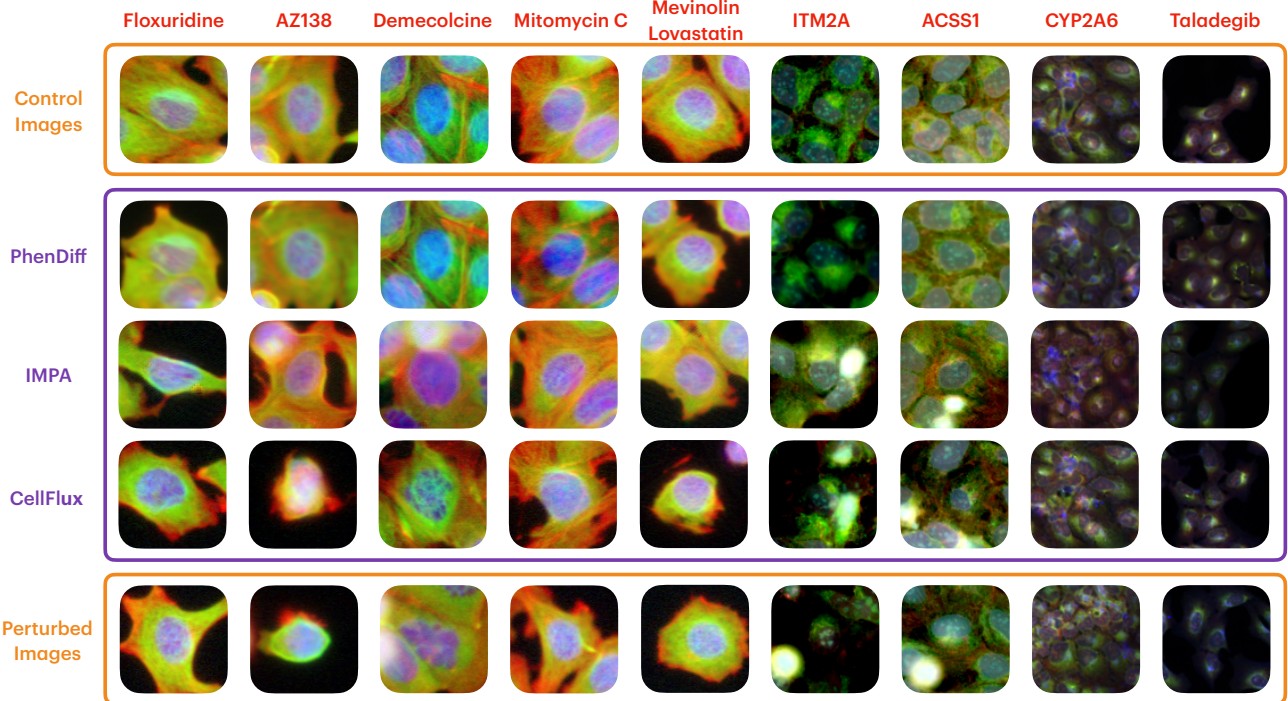

Figure 3. **Qualitative comparisons.** *CellFlux* generates significantly more accurate images that reflect the actual biological effects of perturbations compared to baselines. For example, Floxuridine inhibits DNA replication, leading to reduced cell density; AZ138 is an Eg5 inhibitor, causing cell death and shrinkage; Demecolcine destabilizes microtubules, resulting in smaller, fragmented nuclei. Columns 1–5, 6–7, and 8–9 correspond to samples from the BBBC021, RxRx1, and JUMP datasets, respectively. More drug's mode-of-action in §E.

## 4.3. Main Results

***CellFlux* generates highly realistic cell images.** *CellFlux* outperforms existing methods in capturing cellular morphology across all datasets (Table 1a), achieving overall FID scores of 18.7, 33.0, and 9.0 on BBBC021, RxRx1, and JUMP, respectively — improving FID by 21%–45% compared to previous methods. These gains in both FID and KID metrics confirm that *CellFlux* produces significantly more realistic cell images than prior approaches.

***CellFlux* accurately captures perturbation-specific morphological changes.** As shown in Table 1a, *CellFlux* achieves conditional FID scores of 56.8 (a 26% improvement), 163.5, and 84.4 (a 16% improvement) on BBBC021, RxRx1, and JUMP, respectively. These scores are computed by measuring the distribution distance for each specific perturbation and averaging across all perturbations. Table 1b further highlights *CellFlux*'s performance on six representative chemical and three genetic perturbations. For chemical perturbations, *CellFlux* reduces FID scores by 14–55% compared to prior methods. The smaller improvement (5–12% improvements) on RxRx1 is likely due to the limited number of images per perturbation type.

***CellFlux* preserves biological fidelity across perturbation conditions.** Table 2a presents mode of action (MoA) classification accuracy and F1 on the BBBC021 dataset using generated cell images. MoA describes how a drug affects cellular function and can be inferred from morphology. To assess this, we train an image classifier on real perturbed images and test it on generated ones. *CellFlux* achieves 71.1% MoA accuracy, closely matching real images (72.4%) and significantly surpassing other methods (best: 63.7%), demonstrating its ability to maintain biological fidelity across perturbations. Qualitative comparisons in Figure 3 further highlight *CellFlux*'s accuracy in capturing key biological effects. For example, demecolcine produces smaller, fragmented nuclei, which other methods fail to reproduce accurately.

***CellFlux* generalizes to out-of-distribution (OOD) perturbations.** On BBBC021, *CellFlux* demonstrates strong generalization to novel chemical perturbations never seen during training (Table 2b). It achieves 6%, 28%, and 170% improvements in overall/conditional FID and MoA accuracy over the best baseline. OOD generalization is critical for biological research, enabling the exploration of previously untested interventions and the design of new drugs.

**Ablations highlight the importance of each component in *CellFlux*.** Table 2d shows that removing conditional information, classifier-free guidance, or noise augmentation significantly degrades performance, leading to higher FID scores. These underscore the critical role of each component in enabling *CellFlux*'s state-of-the-art performance.

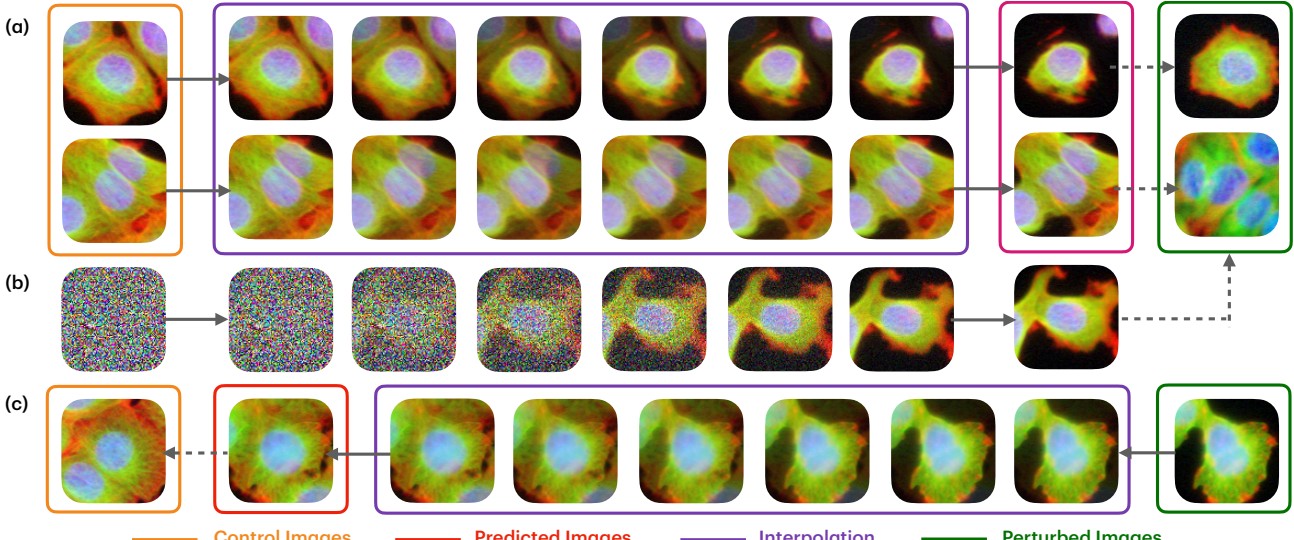

*Figure 4.* **CellFlux** **enables new capabilities.** *(a.1) Batch effect calibration. CellFlux initializes with control images, enabling batch-specific predictions. Comparing predictions from different batches highlights actual perturbation effects (smaller cell size) while filtering out spurious batch effects (cell density variations). (a.2) Interpolation trajectory. CellFlux's learned velocity field supports interpolation between cell states, which might provide insights into the dynamic cell trajectory. (b) Diffusion model comparison. Unlike flow matching, diffusion models that start from noise cannot calibrate batch effects or support interpolation. (c) Reverse trajectory. CellFlux's reversible velocity field can predict prior cell states from perturbed images, offering potential applications such as restoring damaged cells.*

## 4.4. New Capabilities

***CellFlux* addresses batch effects and reveals true perturbation effects.** *CellFlux*'s distribution-to-distribution approach effectively addresses batch effects, a significant challenge in biological experimental data collection. As shown in Figure 4a, when conditioned on two distinct control images with varying cell densities from different batches, *CellFlux* consistently generates the expected perturbation effect (cell shrinkage due to mevinolin) while recapitulating batch-specific artifacts, revealing the true perturbation effect. Table 2c further quantifies the importance of conditioning on the same batch. By comparing generated images conditioned on control images from the same or different batches against the target perturbation images, we find that same-batch conditioning improves conditional FID and MoA accuracy by 14% and 48%. This highlights the importance of modeling control images to more accurately capture true perturbation effects—an aspect often overlooked by prior approaches, such as diffusion models that initialize from noise (Figure 4b).

***CellFlux* has the potential to model cellular morphological change trajectories.** Cell trajectories could offer valuable information about perturbation mechanisms, but capturing them with current imaging technologies remains challenging due to their destructive nature. Since *CellFlux* continuously transforms the source distribution into the target distribution, it can generate smooth interpolation paths between initial and final predicted cell states, producing video-like sequences of cellular transformation based on

given source images (Figure 4a). This suggests a possible approach for simulating morphological trajectories during perturbation response, which diffusion methods cannot achieve (Figure 4b). Additionally, the reversible distribution transformation learned through flow matching enables *CellFlux* to model backward cell state reversion (Figure 4c), which could be useful for studying recovery dynamics and predicting potential treatment outcomes.

## 5. Related Works

**Generative models.** Generative models are a fundamental class of machine learning approaches that learn to model and sample from probability distributions. Traditional methods such as autoregressive models, normalizing flows, and GANs face limitations in generation speed, expressiveness, or training stability (Van Den Oord et al., 2016; Papamakarios et al., 2021; Goodfellow et al., 2014). Recent score-based approaches, particularly diffusion models (Sohl-Dickstein et al., 2015; Song & Ermon, 2019; Ho et al., 2020) and flow matching (Lipman et al., 2023; 2024; Liu et al., 2023; 2024), address these challenges by learning continuous-time transformations between distributions, achieving state-of-the-art performance in generating images, videos, and biological sequences (OpenAI, 2024; Esser et al., 2024; Kondratyuk et al., 2024; Hayes et al., 2025). Unlike diffusion models, which map from Gaussian noise, flow matching directly transforms between arbitrary distributions. This property remains underexplored in machine learning due to limited application scenarios (Liu et al., 2024), yet it

is particularly well-suited for cellular morphology prediction, where accurately modeling the transition from unperturbed to perturbed cell states is crucial.

**Cellular morphology prediction.** Cellular morphology serves as a powerful phenotypic readout in biological research, offering critical insights into cellular states (Perlman et al., 2004; Loo et al., 2007). Predicting morphological changes *in silico* enables rapid virtual drug screening and the development of personalized therapeutic strategies, significantly accelerating biomedical discoveries (Carpenter, 2007; Bunne et al., 2024). While initial progress has been made in this direction, existing approaches face three major limitations. Some neglect control cell images, failing to capture true perturbation changes and making predictions vulnerable to batch effects (Yang et al., 2021; Navidi et al., 2025; Cook et al., 2024). Others rely on outdated generative techniques such as normalizing flows and GANs, which suffer from training instability and limited image fidelity (Lamiable et al., 2023; Palma et al., 2025). Additionally, some methods use suboptimal approaches to model distribution transformation, such as a two-step diffusion process (Bourou et al., 2024; Hung et al., 2024). Our work addresses these challenges by reframing morphology prediction as a distribution-to-distribution translation problem and leveraging flow matching, which naturally models cellular state transformations while ensuring high image quality and stable training, paving the way for constructing virtual cells for biomedical research.

## 6. Conclusion

In this work, we introduce *CellFlux*, a method that leverages flow matching to generate cell images under various perturbations while capturing their trajectories, paving the way for the development of a virtual cell framework for biomedical research. In future work, we plan to scale up *CellFlux* to process terabytes of imaging data encompassing diverse cell types and a wide range of perturbations, enabling the full potential of virtual cell modeling.

## Acknowledgments

This work is partially supported by the Hoffman-Yee Research Grants. E.L. and S.Y. are Chan Zuckerberg Biohub — San Francisco Investigators.

## Impact Statement

*CellFlux* introduces a novel machine learning framework for modeling cellular responses to genetic and chemical perturbations by formulating the task as a distribution-to-distribution transformation and solving it using a principled flow matching approach. This leads to significantly improved predictive performance and unlocks new capabilities such as batch effect correction and perturbation interpolation.

By providing scalable and interpretable computational tools for modeling perturbation responses at both the single-cell and population levels, *CellFlux* addresses critical challenges in experimental biology. It enables rapid in-silico screening of compounds and perturbations, thereby accelerating therapeutic discovery and drug repurposing. In particular, it can guide follow-up experiments toward the most promising candidates, streamlining the drug repurposing pipeline and the search for novel therapeutic targets. In addition to medical applications, *CellFlux* can accelerate basic research into cell biology processes by modeling responses to genetic or chemical perturbations.

However, we acknowledge that these are early attempts to model complex and dynamic biological systems, and future research with larger and more diverse datasets will improve performance. For instance, we are limited by current datasets that focus on a few cancer cell lines, which could introduce bias and may not fully represent normal physiology. Furthermore, while our method enables interpolation between cell states, the biological validity of these interpolations remains unverified; establishing their plausibility will require future work involving ground-truth data and experimental validation.

Despite these limitations, *CellFlux* bridges machine learning and cellular biology, enabling new frontiers in virtual cell modeling, drug discovery, and systems biology research with broad implications for science and medicine.

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

## Summary of Appendix

- §A presents a formal proof supporting our mathematical formulation.

- §B details the *CellFlux* algorithm.

- §C provides additional experimental details.

- §D offers a more in-depth discussion of batch effects.

- §E describes the datasets used in our study.

- §F includes qualitative comparisons of *CellFlux* against baselines.

- §G presents additional visualization of bidirectional trajectories between control images and perturbed images.

- §H provides additional results comparing our method to baselines.

- §I provides a table to compare our work with related works.

- §J discusses future directions for validating the interpolation trajectories generated by *CellFlux*.

# A. Theory Proof

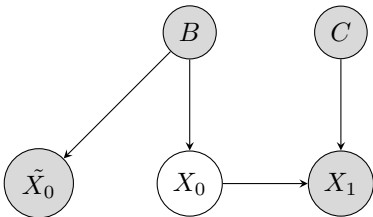

**Proposition 1.** *Given random variables $B$, $C$, $X_0$, $\tilde{X}_0$, and $X_1$ following the graphical model above with joint distribution $p(b, c, x_0, \tilde{x}_0, x_1)$, the distribution $p(x_1|x_0, c)$ can be better approximated by the conditional distribution $p(x_1|\tilde{x}_0, c)$ than $p(x_1|c)$ in expectation. Formally,*

$$\mathbb{E}_{p(x_0, \tilde{x}_0, c)}\left[D_{KL}(p(x_1|x_0, c)||p(x_1|\tilde{x}_0, c))\right] \leq \mathbb{E}_{p(x_0, c)}\left[D_{KL}(p(x_1|x_0, c)||p(x_1|c))\right]$$

*Proof.* According to the definition of conditional mutual information, the term on the right-hand side can be expressed as:

$$\begin{aligned}
\mathbb{E}_{p(x_0, c)}\left[D_{\text{KL}}(p(x_1|x_0, c)||p(x_1|c))\right] &= \int p(x_0, c)p(x_1|x_0, c)\log\frac{p(x_1|x_0, c)}{p(x_1|c)}dx_1 dx_0 dc \\
&= \int p(c)\mathbb{E}_{p(x_0, x_1|c)}\left[\log\frac{p(x_1, x_0|c)}{p(x_1|c)p(x_0|c)}\right]dc \\
&= \mathbb{E}_{p(c)}\left[D_{\text{KL}}(p(x_1, x_0|c)||p(x_1|c)p(x_0|c))\right] = I(X_1; X_0|C)
\end{aligned}$$

Based on the graphical model, we have the conditional independence $X_1 \perp\!\!\!\perp \tilde{X}_0 | X_0, C$. Thus, we have $p(x_1|x_0, c) = p(x_1|x_0, \tilde{x}_0, c)$. Similarly, we can express the term on the right-hand side as conditional mutual information:

$$\begin{aligned}
\mathbb{E}_{p(x_0, \tilde{x}_0, c)}\left[D_{\text{KL}}(p(x_1|x_0, c)||p(x_1|\tilde{x}_0, c))\right] &= \mathbb{E}_{p(x_0, \tilde{x}_0, c)}\left[D_{\text{KL}}(p(x_1|x_0, \tilde{x}_0, c)||p(x_1|\tilde{x}_0, c))\right] \\
&= \int p(x_0, \tilde{x}_0, c)p(x_1|x_0, \tilde{x}_0, c)\log\frac{p(x_1|x_0, \tilde{x}_0, c)}{p(x_1|\tilde{x}_0, c)}dx_1 d\tilde{x}_0 dx_0 dc \\
&= I(X_1; X_0|\tilde{X}_0, C)
\end{aligned}$$

Further, based on the property of conditional mutual information, we have

$$\begin{aligned}
I(X_1; X_0|C) &= I(X_1; X_0|\tilde{X}_0, C) + I(X_1; \tilde{X}_0|C) - I(X_1; \tilde{X}_0|X_0, C) \\
&= I(X_1; X_0|\tilde{X}_0, C) + I(X_1; \tilde{X}_0|C)
\end{aligned}$$

where the second equality is due to the conditional independence relationship $X_1 \perp\!\!\!\perp \tilde{X}_0 | X_0, C$, and $I(X_1; \tilde{X}_0|X_0, C) = 0$

Therefore,

$$\begin{aligned}
\mathbb{E}_{p(x_0, c)}\left[D_{\text{KL}}(p(x_1|x_0, c)||p(x_1|c))\right] &= \mathbb{E}_{p(x_0, \tilde{x}_0, c)}\left[D_{\text{KL}}(p(x_1|x_0, c)||p(x_1|\tilde{x}_0, c))\right] + I(X_1; \tilde{X}_0|C) \\
&\geq \mathbb{E}_{p(x_0, \tilde{x}_0, c)}\left[D_{\text{KL}}(p(x_1|x_0, c)||p(x_1|\tilde{x}_0, c))\right]
\end{aligned}$$

The inequality holds strictly when $I(X_1; \tilde{X}_0|C) > 0$, i.e., $X_1 \not\perp\!\!\!\perp \tilde{X}_0|C$, which generally holds true when batch effect exists and variables $X_0$ and $\tilde{X}_0$ are associated by $B$ according to the graphical model.

$\square$

# B. *CellFlux* Algorithm

---

**Algorithm 1** *CellFlux* Algorithm

---

**Training Process:**

**input** Initial distribution $p_0$, target distribution $p_1$, perturbation $c$, neural network $v_\theta(x_t, t, c)$, noise injection probability $p_n$, condition drop probability $p_c$, learning rate $\eta$, number of iterations $N$

**output** Trained neural network $v_\theta$

    **for** each iteration $i = 1, \ldots, N$ **do**

        Sample $x_0 \sim p_0$ and $x_1 \sim p_1$

        Sample $t \sim \text{Uniform}[0, 1]$

        Inject noise $x_0 \leftarrow x_0 + \epsilon$, $\epsilon \sim \mathcal{N}(0, I)$ with $p_n$

        Drop condition $c \leftarrow \phi$ with $p_c$

        Interpolate $x_t \leftarrow tx_1 + (1 - t)x_0$

        Compute true velocity $v(x_t, t, c) \leftarrow x_1 - x_0$

        Predict velocity using neural network $v_\theta(x_t, t, c)$

        Compute loss $\mathcal{L} \leftarrow \|v_\theta(x_t, t, c) - v(x_t, t, c)\|_2^2$

        Update $\theta$ using gradient descent $\theta \leftarrow \theta - \eta\nabla_\theta\mathcal{L}$

    **end for**

**Inference Process:**

**input** Initial sample $x_0 \sim p_0$, perturbation $c$, step size $\Delta t$, classifier-free guidance strength $\alpha$

**output** Generated sample $x_1 \sim p_1$

    Initialize $x_t \leftarrow x_0$

    **for** $t = 0$ to 1 with step size $\Delta t$ **do**

        Computer velocity with classifier-free guidance $v_\theta^{\text{CFG}}(x_t, t, c) \leftarrow \alpha v_\theta(x_t, t, c) + (1 - \alpha)v_\theta(x_t, t, \emptyset)$

        Update $x_t \leftarrow x_t + \Delta t \cdot v_\theta^{\text{CFG}}(x_t, t, c)$

    **end for**

    Output final state $x_1 \leftarrow x_t$

---

Algorithm 1 provides a detailed overview of the *CellFlux* algorithm, covering both training and inference. During training, the model learns to predict velocity between an initial and target distribution by interpolating between samples, applying noise and condition dropout, and optimizing an L2 loss between predicted and true velocities. In inference, the trained model iteratively updates a sample from the initial distribution toward the target distribution using classifier-free guidance, ultimately generating a new sample that aligns with the target distribution.

## C. Experimental Details

**Model architecture.** *CellFlux* employs a UNet-based velocity field parameterization with input and output channels matching the dataset. It features four stages for downsampling and upsampling, with each stage halving or doubling the resolution and using a hidden size of 128. This hierarchical UNet design focuses on efficient 2D spatial learning for pixel-level flow matching.

**Perturbation encoding.** We encode perturbations following IMPA's approach (Palma et al., 2025). For chemical embeddings, we use 1024-dimensional Morgan Fingerprints generated with RDKit. For gene embeddings, CRISPR and ORF embeddings combine Gene2Vec with HyenaDNA-derived sequence representations, resulting in final dimensions of 328 and 456, respectively.

**Training details.** Models are trained for 100 epochs on 4 A100 GPUs using the Adam optimizer with a learning rate of 1e-4 and a batch size of 128, requiring 8, 16, and 36 hours for BBBC021, RxRx1, and JUMP, respectively. The noise injection probability, condition drop probability, and classifier-free guidance strength are set to 0.5, 0.2, and 1.2, respectively. Models are selected based on the lowest FID scores on the validation set.

# D. Batch Effects

## 1. What Are Batch Effects in Microscopy Experiments?

Batch effects refer to a form of *distribution shift* in microscopy experiments, where non-biological variations arise due to differences in experimental conditions across imaging sessions or batches. These effects can be classified as a type of *covariate shift*, where technical factors alter the distribution of input features, including:

- **Microscope or camera settings** – Variations in sensor sensitivity, illumination, resolution, or imaging modalities.
- **Experimental procedures** – Differences in sample preparation, staining protocols, reagent batches, or handling by different researchers.
- **Environmental conditions** – Changes in temperature, humidity, or laboratory-specific conditions that may be difficult to control.

As a result, images of biologically identical cells may appear different solely due to variations in imaging conditions rather than biological differences.

## 2. Why Do Batch Effects Matter?

Batch effects pose a major challenge to reproducible biomedical research by obscuring true biological effects of perturbations. Additionally, machine learning models may inadvertently learn batch-specific artifacts instead of meaningful biological patterns. Key issues caused by batch effects include:

- **Poor generalization** – Models trained on batch-affected images may fail to classify new samples from a different experimental setup.
- **False discoveries** – Uncorrected batch effects can confound biological signals, leading to misleading conclusions.
- **Reduced reproducibility** – Results may not replicate across laboratories or imaging systems due to unaccounted technical biases.

## 3. Visualization of Batch Effects

Figure 5 visualizes three batches of BBBC021 images using PCA, showing that each batch forms a distinct cluster. Notably, control (ctrl) and perturbed (trt) images from the same batch cluster together, rather than forming separate control and target clusters. This illustrates the **batch effect**—a systematic bias within each batch that is unrelated to the perturbation itself.

## 4. How *CellFlux* Addresses Batch Effects?

*CellFlux* mitigates batch effects by using control images as initialization during both training and inference, transporting them to target images within the same batch. This ensures that the model learns only the **relative difference** between control and perturbed images. By conditioning on control images from different batches, *CellFlux* effectively captures the **true perturbation effect** while preserving batch-specific artifacts. Figure 5 demonstrates this, showing that predicted images remain within the same batch cluster when given a control image from that cluster.

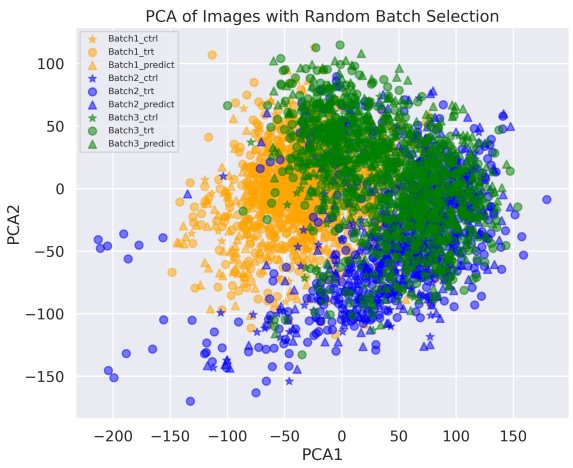

*Figure 5.* **Visualization of batch effects in BBBC021 and how *CellFlux* addresses batch effects.**

# E. Datasets

As described below, all data used in this study are publicly available and utilized under their respective licenses. No new data were generated for this study.

**BBBC021 dataset.** We utilized the BBBC021v1 image set (Caie et al., 2010), available from the Broad Bioimage Benchmark Collection (Ljosa et al., 2012). The BBBC021 dataset focuses on chemical perturbations in MCF-7 breast cancer cells, serving as a robust benchmark for image-based phenotypic profiling. It comprises 97,504 fluorescent microscopy images of cells treated with 113 small molecules across eight concentrations, targeting diverse cellular mechanisms such as actin disruption, Aurora kinase inhibition, and microtubule stabilization. Each image includes multi-channel labels for DNA, F-actin, and beta-tubulin, facilitating detailed morphological analysis. Metadata provides mechanism-of-action (MOA) annotations for compounds and experimental conditions, enabling applications in mechanistic prediction and phenotypic similarity analysis. Table 3 shows MoA classes for all BBBC021 perturbations. Images were processed at a resolution suitable for segmentation and deep learning tasks.

**RxRx1 dataset.** The RxRx1 dataset (Sypetkowski et al., 2023), available under a CC-BY-NC-SA-4.0 license from Recursion Pharmaceuticals at rxrx.ai, focuses on genetic perturbations using CRISPR-mediated gene knockouts. It contains 170,943 images representing 1,042 genetic perturbations in HUVEC cells, with control conditions to address experimental variability. Images were captured across six channels, including nuclear and cytoskeletal markers, enabling high-dimensional phenotypic analysis. Preprocessing steps included segmentation, cropping, and resizing to standardize the data for computational analysis. This dataset supports tasks such as feature extraction, phenotypic clustering, and representation learning.

**JUMP dataset (CPJUMP1).** The JUMP dataset (Chandrasekaran et al., 2023), available under a CC0 1.0 license, integrates both genetic and chemical perturbations, offering the most comprehensive image-based profiling resource to date. It includes approximately 3 million images capturing the phenotypic responses of 75 million single cells to genetic knockouts (CRISPR/ORF) and chemical perturbations. Key features include:

- **Chemical-genetic pairing:** Perturbations targeting the same genes are tested in parallel to assess phenotypic convergence or divergence.

- **Controlled conditions:** Imaging was standardized across cell types (U2OS and A549), time points (short and extended durations), and experimental setups.

- **Primary group:** Forty plates profiling CRISPR knockouts and ORF overexpression.

- **Secondary group:** Additional plates exploring extended experimental conditions.

The JUMP dataset uniquely enables the study of phenotypic relationships between genetic and chemical perturbations and supports the development of predictive models for multi-modal cellular responses. Public access to the dataset and associated analysis pipelines is available via Broad's JUMP repository.

| Compound | MoA |
|---|---|
| Cytochalasin B | Actin disruptors |
| Cytochalasin D | Actin disruptors |
| Latrunculin B | Actin disruptors |
| AZ258 | Aurora kinase inhibitors |
| AZ841 | Aurora kinase inhibitors |
| Mevinolin/Lovastatin | Cholesterol-lowering |
| Simvastatin | Cholesterol-lowering |
| Chlorambucil | DNA damage |
| Cisplatin | DNA damage |
| Etoposide | DNA damage |
| Mitomycin C | DNA damage |
| Camptothecin | DNA replication |
| Floxuridine | DNA replication |
| Methotrexate | DNA replication |
| Mitoxantrone | DNA replication |
| AZ138 | Eg5 inhibitors |
| PP-2 | Epithelial |
| Alsterpaullone | Kinase inhibitors |
| Bryostatin | Kinase inhibitors |
| PD-169316 | Kinase inhibitors |
| Colchicine | Microtubule destabilizers |
| Demecolcine | Microtubule destabilizers |
| Nocodazole | Microtubule destabilizers |
| Vincristine | Microtubule destabilizers |
| Docetaxel | Microtubule stabilizers |
| Epothilone B | Microtubule stabilizers |
| Taxol | Microtubule stabilizers |
| ALLN | Protein degradation |
| Lactacystin | Protein degradation |
| MG-132 | Protein degradation |
| Proteasome inhibitor I | Protein degradation |
| Anisomycin | Protein synthesis |
| Cyclohexamide | Protein synthesis |
| Emetine | Protein synthesis |
| DMSO | DMSO |

*Table 3.* **Modes of action (MoA) for compounds in BBBC021.**

## F. Qualitative Comparison

In this section, we present additional generated samples to further demonstrate the effectiveness of our method. Figures 6, 7, and 8 show qualitative comparisons on the BBBC021, RxRx1, and JUMP datasets, respectively. Our approach more accurately captures key biological effects, whereas images generated by IMPA fail to reflect real biological responses, and those from PhenDiff appear blurry with significant detail loss.

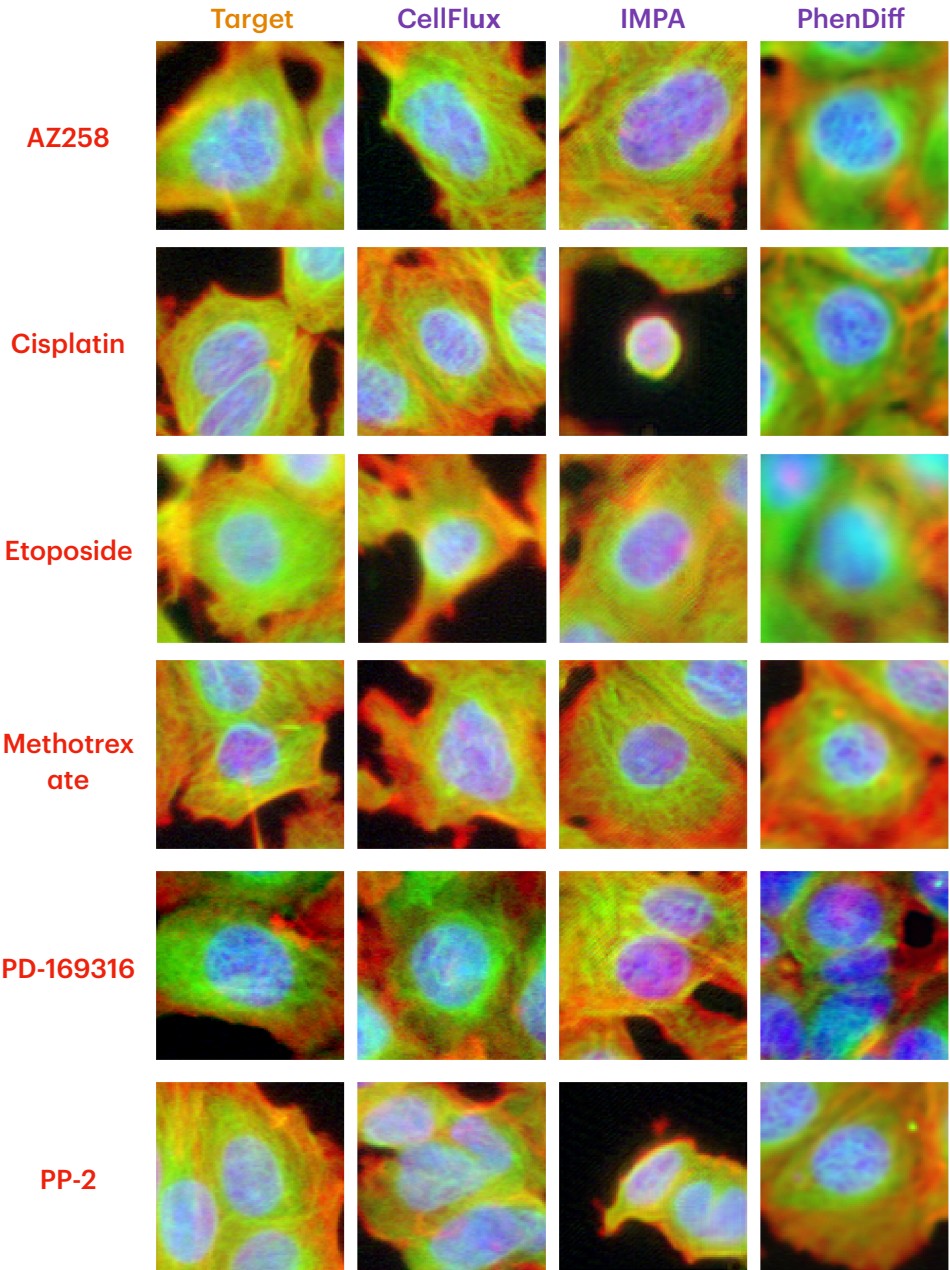

*Figure 6.* **More qualitative comparisons of generated samples on BBBC021.**

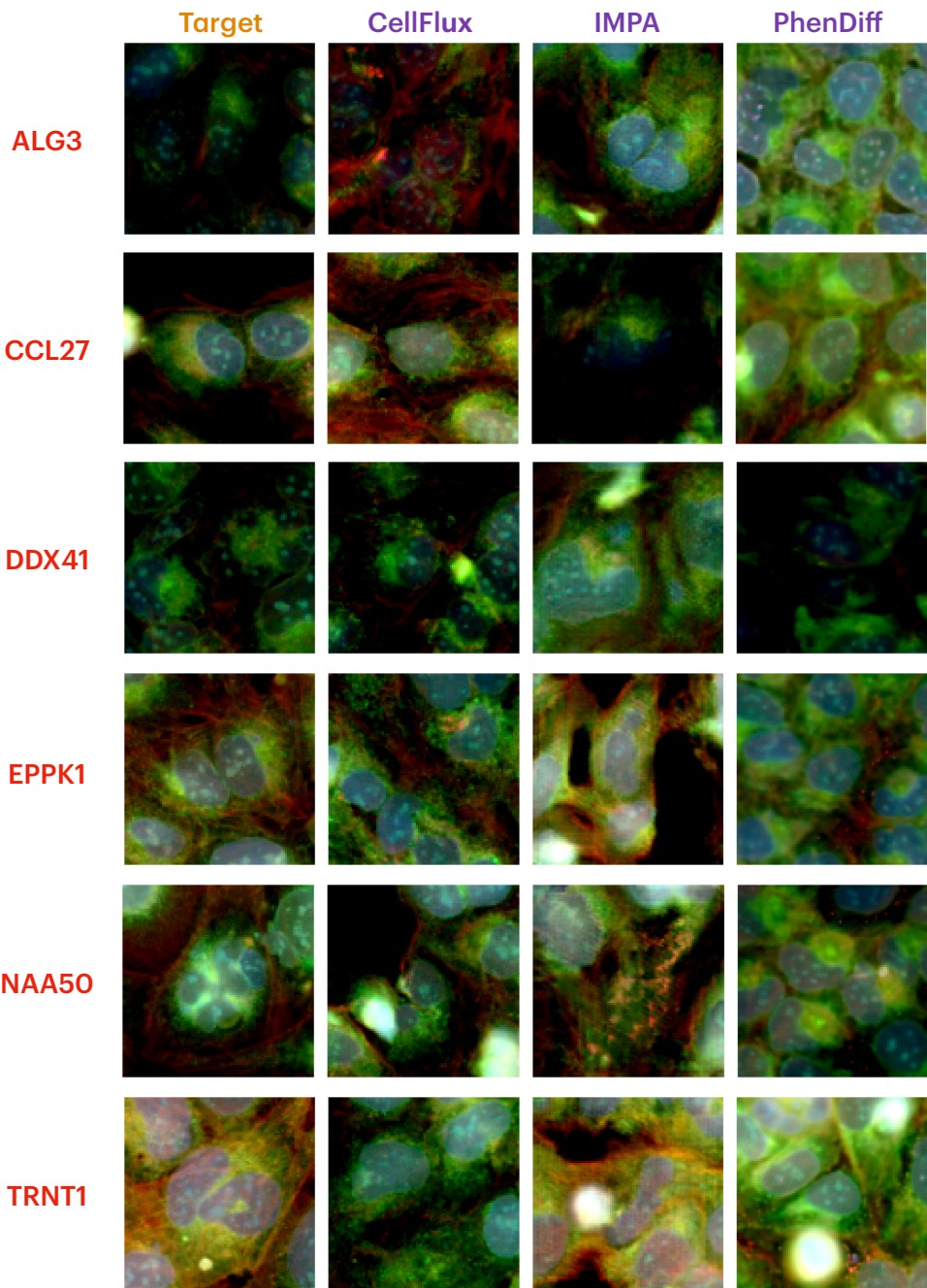

*Figure 7.* **More qualitative comparisons of generated samples on RxRx1.**

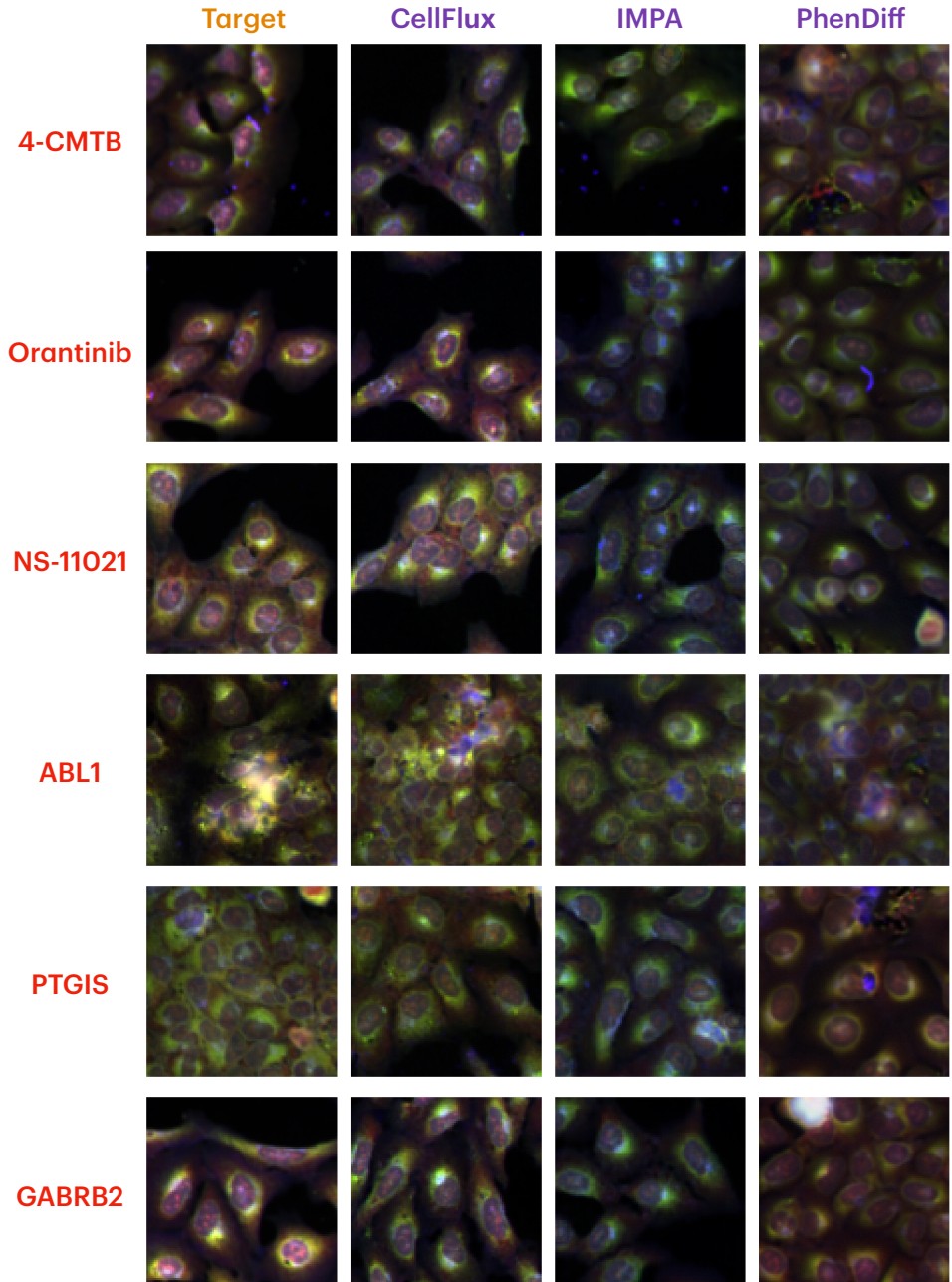

Figure 8. **More qualitative comparisons of generated samples on JUMP.**

# G. Trajectory

**Forward interpolation.** Our generation process aims to transform a control image into its corresponding perturbed image using our flow matching model. This is achieved by iteratively solving an ODE, where the velocity field predicted by the model guides the transformation at each timestep. As iterations progress, the image gradually evolves towards its final state at $t = 1$, representing the fully perturbed cell morphology.

**Backward interpolation.** Due to the bidirectional nature of our model, we can also perform a reversible generation process by inverting the velocity direction. This allows us to start from the perturbed image and gradually recover the original control image, demonstrating the reversible capabilities of our method.

**Trajectory examples.** Figures 9 and 10 illustrate these bidirectional transformations. The top section of each figure depicts the forward trajectory, where the control image is progressively updated based on the learned velocity field, ultimately generating the perturbed image at $t = 1$. The bottom section shows the reverse trajectory, where the process is reversed, progressively reconstructing the original control image. This capability, which is absent in diffusion-based methods, offers a promising approach for simulating morphological trajectories during perturbation responses. Moreover, *CellFlux*'s reversible distribution transformation enables modeling of backward transitions in cell states, with potential applications in studying recovery dynamics and predicting treatment outcomes.

To further demonstrate our approach, we present trajectory examples for two drugs. The first, PP-2, reduces cell adhesion and disrupts actin reorganization, leading to a more dispersed cell distribution. In Figure 9, the forward trajectory shows cells transitioning from a clustered to a more diffuse state, while the reverse trajectory restores the original aggregation. The second, Chlorambucil, induces pyknosis (nuclear shrinkage). In Figure 10, the forward process shows one of the three nuclei undergoing cell death or division, leaving only two nuclei in the final state, while the reverse trajectory reconstructs the original three-nucleus configuration. These results highlight our method's ability to capture biologically meaningful morphological transitions in both directions.

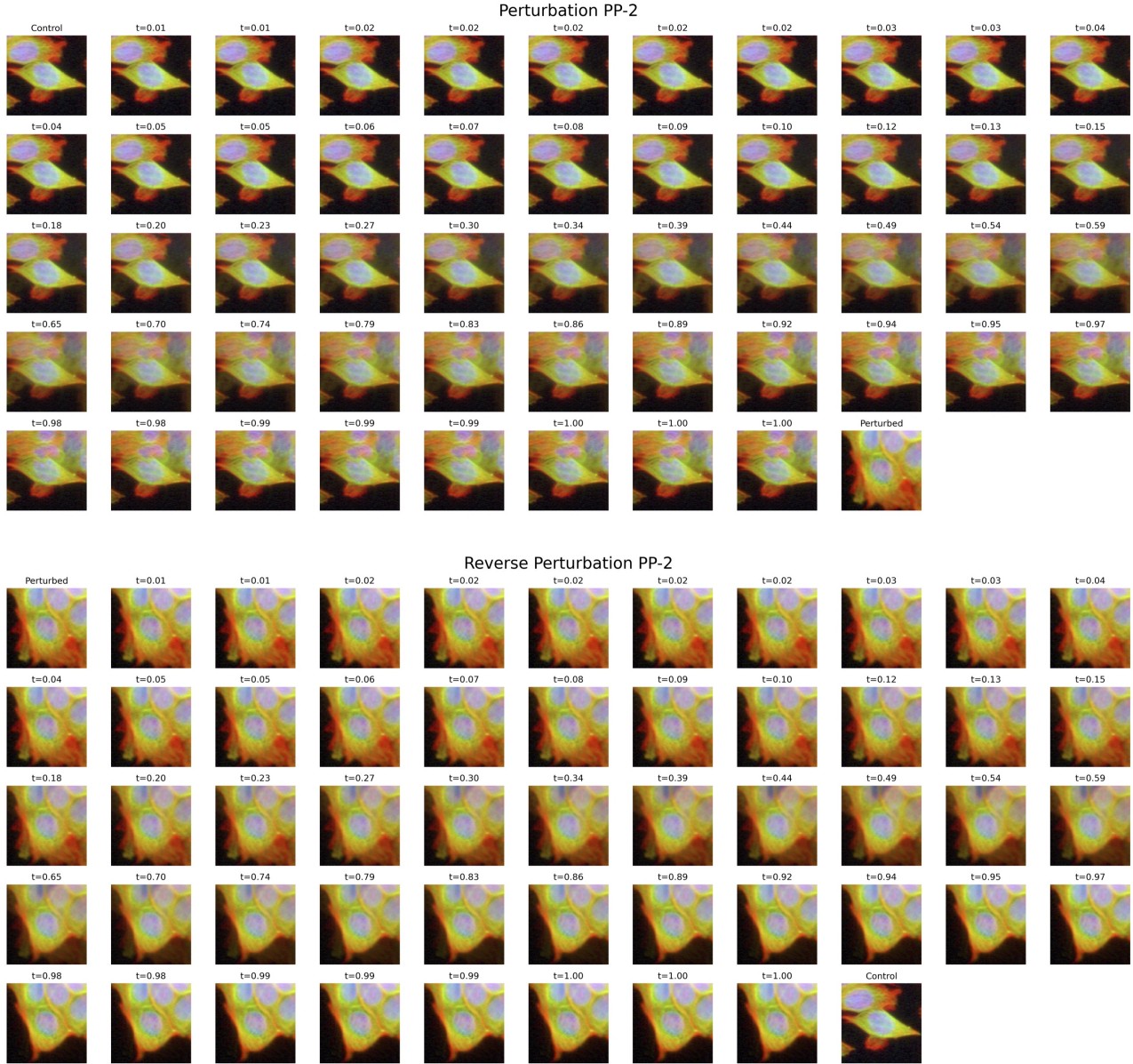

*Figure 9.* **(1/2) Bidirectional interpolation trajectory in BBBC021.**

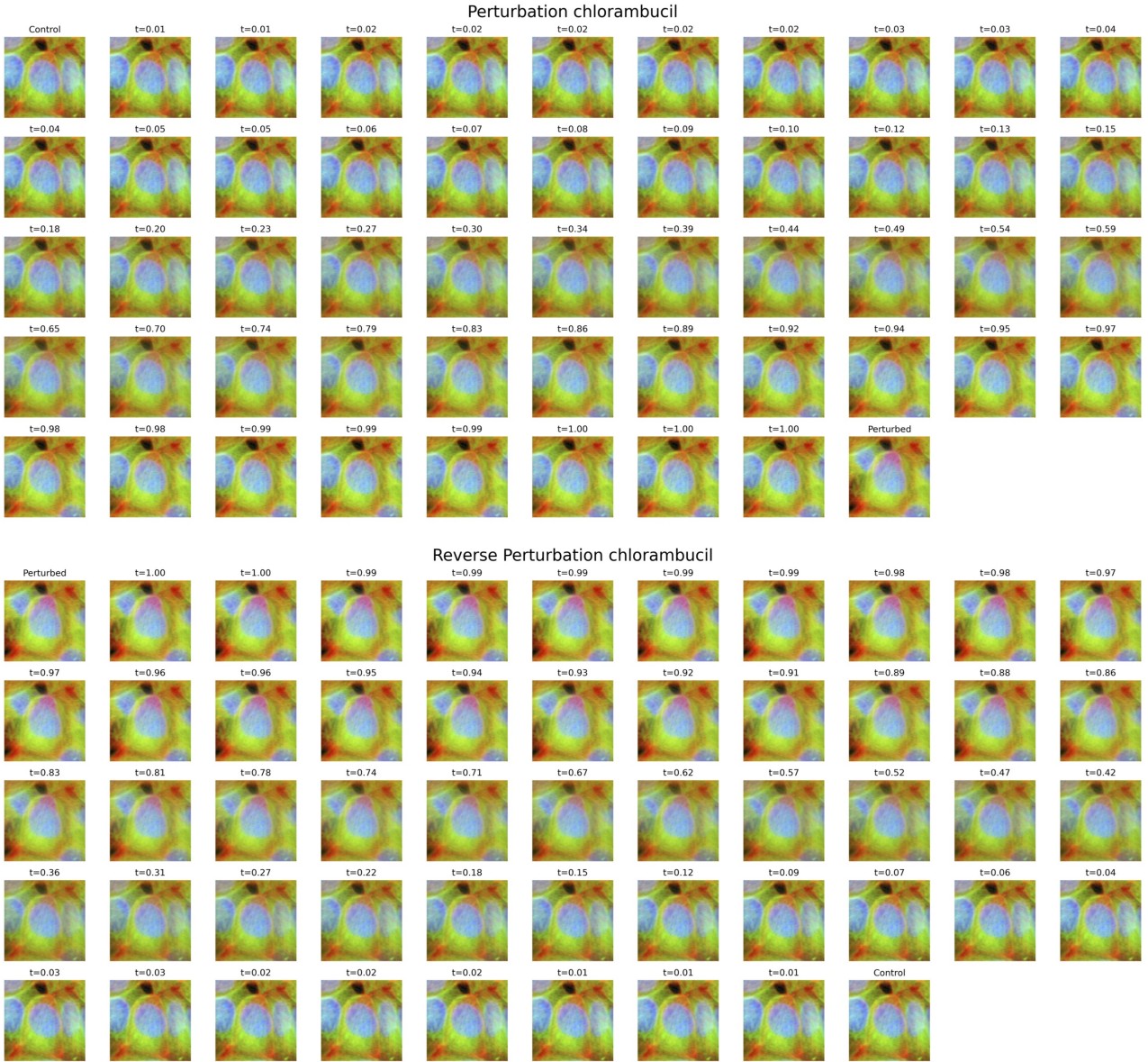

*Figure 10.* **(2/2) Bidirectional interpolation trajectory in BBBC021.**

## H. More Results

**Out-of-distribution generalization.** Table 4 reports results on the out-of-distribution (OOD) set in BBBC021, evaluating performance on perturbations absent from the training set. This highlights our method's strong generalization ability to novel chemical perturbations. The FID score measures the similarity between generated and real distributions, with lower values indicating a closer match. As shown in the table, our method effectively captures the biological effects of each perturbation, generating images that closely resemble real cellular responses. Robust OOD generalization is essential for biological research, enabling the exploration of untested interventions, analysis of unknown cellular responses, and the design of new drugs by simulating effects before experimental validation.

| Method | AZ841 | Cyclohexamide | Cytochalasin D | Docetaxel | Epothilone B | Lactacystin | Latrunculin B | Simvastatin |
|---|---|---|---|---|---|---|---|---|
| PhenDiff | 136.5 | 224.0 | 180.3 | 160.1 | 131.5 | 139.7 | 132.5 | 108.9 |
| IMPA | 131.7 | 189.9 | 180.6 | 130.6 | 120.7 | 133.7 | 128.5 | **79.7** |
| CellFlux | **84.1** | **99.5** | **129.4** | **81.9** | **93.7** | **106.9** | **97.9** | 90.9 |

Table 4. **Out-of-distribution generalization results per perturbation.**

**Effect of sample size on FID/KID.** FID and KID are known to be sensitive to the number of samples. We evaluate performance across varying sample sizes on BBBC021 (1K–5K, limited to 6K test images) and JUMP (10K–20K). As shown in Table 5, *CellFlux* consistently outperforms all baselines across all sample sizes, achieving 30–45% relative improvement and demonstrating the robustness of its improvement regardless of sample size.

| Method | 1K FID | 2.5K FID | 5K FID | 10K FID | 20K FID | 1K KID | 2.5K KID | 5K KID | 10K KID | 20K KID |
|---|---|---|---|---|---|---|---|---|---|---|
| PhenDiff | 71.3 | 64.3 | 49.5 | 47.5 | 46.1 | 2.55 | 3.68 | 3.10 | 4.95 | 5.09 |
| IMPA | 52.4 | 41.4 | 33.7 | 14.0 | 12.9 | 3.20 | 3.38 | 2.60 | 1.04 | 1.05 |
| CellFlux | **34.7** | **25.2** | **18.7** | **8.5** | **7.5** | **1.67** | **1.90** | **1.62** | **0.63** | **0.63** |

Table 5. **FID and KID across different sample sizes.**

**Comparison with more baselines.** Cell morphology prediction is a new task with only six baselines (Table 8). We included the only two published methods using control images (Bourou et al., 2024; Palma et al., 2025); others are unpublished (Hung et al., 2024; Cook et al., 2024), lack code (Yang et al., 2021; Cook et al., 2024), or omit controls (Yang et al., 2021; Navidi et al., 2025; Cook et al., 2024). We further compared MorphoDiff (Navidi et al., 2025), a recent diffusion-based method, without using control images. Under our setup on BBBC021, *CellFlux* outperforms it in image quality and MoA metrics.

| Method | $FID_o$ | $FID_c$ | $KID_o$ | $KID_c$ | MoA Accuracy | MoA Macro-F1 | MoA Weighted-F1 |
|---|---|---|---|---|---|---|---|
| MorphoDiff | 65.8 | 114.1 | 7.99 | 7.97 | 38.3 | 24.5 | 34.2 |
| CellFlux | **18.7** | **56.8** | **1.62** | **1.59** | **71.2** | **49.0** | **70.7** |

Table 6. *CellFlux* **outperforms MorphoDiff on BBBC021 in both image quality and MoA classification metrics.**

**Cross-dataset transfer.** To assess whether the learned model can generalize to highly out-of-distribution settings, we conduct a cross-dataset transfer experiment by applying a *CellFlux* model trained on BBBC021 to RxRx1 and JUMP images. Surprisingly, we observe that *CellFlux* successfully transfers and applies perturbation effects despite substantial domain shifts (Figure 11), highlighting its potential as a unified foundation model across diverse perturbation datasets. Note that there are no target images for RxRx1 and JUMP, as these datasets do not contain the corresponding perturbation conditions.

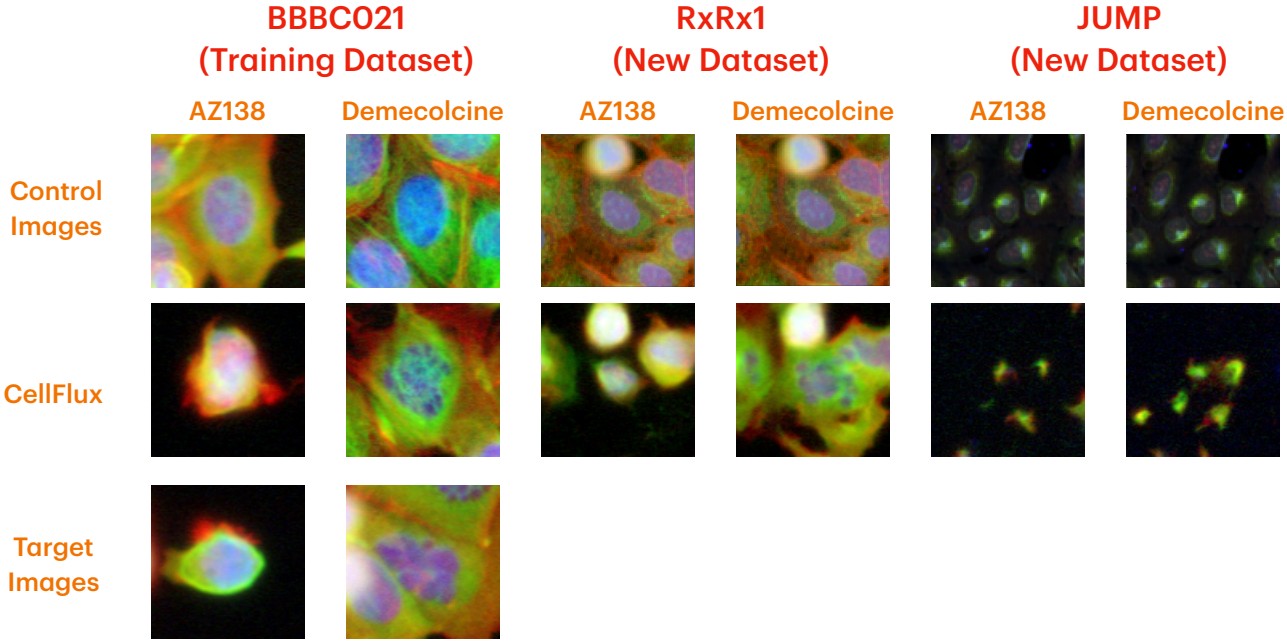

*Figure 11.* **Cross-dataset transfer of *CellFlux*.** Although *CellFlux* is trained solely on BBBC021, it demonstrates zero-shot generalization to two unseen datasets—RxRx1 and JUMP. Notably, it can predict morphological changes induced by perturbations (AZ138 and Demecolcine) that are absent from both datasets. AZ138, an Eg5 inhibitor, leads to cell shrinkage and death, while Demecolcine disrupts microtubules, resulting in smaller, fragmented nuclei.

**CellProfiler metrics.** To further evaluate whether *CellFlux* can capture perturbation-specific morphological changes, we extracted CellProfiler features related to nuclear size under three perturbations known to enlarge nuclei (taxol, vincristine, and demecolcine) using the BBBC021 dataset. As shown in Table 7 (mean and 95% confidence interval reported), *CellFlux* most closely matches the real perturbed morphology in terms of nuclear size.

| Method | taxol | vincristine | demecolcine |
|---|---|---|---|
| Control | 1612.0 ± 39.5 | 1612.0 ± 39.5 | 1612.0 ± 39.5 |
| Target | 2296.7 ± 190.1 | 2365.5 ± 125.5 | 2311.0 ± 136.1 |
| PhenDiff | 1755.9 ± 138.2 | 1947.8 ± 70.8 | 2118.8 ± 102.7 |
| IMPA | 2088.3 ± 190.8 | 2116.9 ± 107.1 | 2386.5 ± 123.9 |
| CellFlux | **2141.0 ± 166.6** | **2276.4 ± 115.6** | **2323.8 ± 121.9** |

*Table 7.* **Comparison of CellProfiler nuclear size features under three compounds known to enlarge nuclei.**

# I. Related Works

Table 8 presents a brief comparison of our work with existing methods for cellular morphology generation.

| Paper | Generative Model | Use Control Image | How to Use Control Image |
|---|---|---|---|
| Mol2Image (Yang et al., 2021) | Normalizing Flows | No | - |
| PhenDiff (Bourou et al., 2024) | Diffusion | Yes | Map control to noise then to target |
| LUMIC (Hung et al., 2024) | Diffusion | Yes | Add DINO embedding as condition |
| pDIFF (Cook et al., 2024) | Diffusion | No | - |
| MorphoDiff (Navidi et al., 2025) | Diffusion | No | - |
| IMPA (Palma et al., 2025) | GAN | Yes | Add AdaIn layers to GAN |
| CellFlux (Ours) | Flow Matching | Yes | Initialization as source distribution |

*Table 8.* **Related works on cell morphology generation.**

## J. Biological Validation of Interpolation Trajectories

While *CellFlux* enables interpolation of perturbation trajectories, the biological relevance of the interpolated states remains unverified. Validating these trajectories is a key next step. Although existing datasets lack ground-truth labels for intermediate states, future work could explore the following directions:

- **Dose interpolation:** Some datasets include images under multiple dosage levels. We can test whether interpolation from control to high dose yields intermediate states that resemble realistic medium-dose morphologies.

- **Timepoint interpolation:** For datasets with multiple timepoints (e.g., day 0, day 5, day 10), we can evaluate whether interpolation from control to day 10 recovers morphologies consistent with intermediate timepoints such as day 5.

- **Drug perturbation interpolation:** Current datasets rarely include fine-grained trajectories post-treatment. Validating such interpolations may require collecting new wet-lab data, such as live-cell imaging over time.

In addition to validation, future work could improve the biological plausibility of interpolation trajectories and avoid degenerate "shortest path in pixel space" artifacts by:

- **Interpolating in latent space:** Performing interpolation in a learned latent space (e.g., via an autoencoder) rather than pixel space may encourage trajectories to follow a more structured biological manifold.

- **Adding supervision from intermediate states:** When datasets contain known intermediate states (e.g., medium dose), models can be trained to explicitly pass through these points during interpolation.

- **Adding constraints:** Incorporating additional constraints—such as a GAN loss—can guide interpolated images to resemble real cell images, improving biological realism.

