# OpenReview forum: "CellFlux: Simulating Cellular Morphology Changes via Flow Matching"
_ICML.cc/2025/Conference — ICML 2025 poster_

### Official Review · Reviewer_L3Zf · 2025-02-25

**Overall Recommendation:** 4

**Summary:**

This paper proposes CellFlow, a generative model for cell microscopy images in the presence of chemical and/or genetic perturbations. In contrast to existing methods that tackle this problem, CellFlow can explicitly take into account batch effects by learning a distribution-level mapping between unperturbed (control) images and perturbed images within the same experimental batch. This is achieved by using flow matching as a tool for unsupervised image translation. The experimental results demonstrate the importance of taking batch effects into account, and show improved image quality and generalization across the different perturbations in the data.

## update after rebuttal

Recommendation increased during rebuttal phase.

**Claims And Evidence:**

The claims made are as follows:

1. CellFlow models distribution-wise mappings from unperturbed to perturbed images and consequently distinguishes perturbation effects from batch effects.

2. CellFlow features improved image quality on three different genetic and/or chemical perturbation datasets over two baseline models.

3. CellFlow generalizes to perturbations not seen during training.

4. CellFlow enables continuous interpolation between unperturbed and perturbed cellular states, offering a means to study the perturbation dynamics.

Overall, all claims are supported by experimental results. However, see also the comments in the experimental design or analyses section of the review.

**Essential References Not Discussed:**

Classifier-Free guidance for Flow Matching has been introduced in [3], and it would be good to add the citation in Section 3.3.

[3] Guided Flows for Generative Modeling and Decision Making. https://arxiv.org/abs/2311.13443

**Experimental Designs Or Analyses:**

1. I assume FID and KID scores are calculated using a model trained on ImageNet. Can we be sure that those scores are meaningful in the context of cell images, as such images are not aligned with images of natural scenes on which the model was trained?

2. It is claimed that CellFlow provides a potential means to study perturbation trajectories. However, from 4.4 it remains unclear whether or not these trajectories correspond to something that is biologically meaningful. For example, in the top row of Fig. 4a, it seems that the model has learned to slowly dim the pixel values that are relatively far from the nucleus, which makes sense since this would roughly correspond to the shortest path between the unperturbed and perturbed images in pixel space. However, this is not necessarily a biologically plausible path.

3. In addition to ML-enabled metrics like FID score and MoA, it could be beneficial to also add biologically relevant metrics (e.g. nucleus size) that describe the morphology of the cell, and see whether these match between the generated and true perturbed cell images.

**Methods And Evaluation Criteria:**

The datasets and methods used appear suitable for the problem statement of this paper. In particular, flow matching's capabilities to map arbitrary source distributions to the target distribution aligns well with the paper's goal of generating perturbed cellular morphologies, starting from unperturbed cells of the same experimental batch.

**Other Comments Or Suggestions:**

N/A

**Other Strengths And Weaknesses:**

Strengths:

1. The paper addresses a relevant application and is well-written.

2. See the points under 'methods and evaluation criteria'



Weaknesses:

1. See the comments under experimental design and analyses for potential weaknesses.

**Questions For Authors:**

Please respond to the comments under experimental design or analyses in particular.

**Relation To Broader Scientific Literature:**

The paper uses the image-to-image translation capabilities of flow matching (earlier explored in e.g. [1] and [2]) to solve the problem of mapping the distribution of unperturbed cell images to the distribution of perturbed cell images. This approach is shown to improve over baseline models that attempt to model the effect of perturbations on cell morphology. So, although both the method and problem are not novel by themselves, to my knowledge this is (1) the first application of flow matching to the problem, and (2) the technology enables taking batch effects into account, which prior methods for the application could not, and the results demonstrate the benefits of doing so.

[1] Improving and generalizing flow-based generative models with minibatch optimal transport. https://arxiv.org/abs/2302.00482

[2] Flow Straight and Fast: Learning to Generate and Transfer Data with Rectified Flow. https://arxiv.org/abs/2209.03003

**Theoretical Claims:**

I read the proof of proposition 1 in Appendix A and did not notice any issues, but I did not thoroughly check its correctness.

---

> ### Author Rebuttal · Authors · 2025-04-01
>
> We thank Reviewer L3Zf for their thoughtful comments and for recognizing the paper’s well-supported claims, appropriate methodology and evaluation, sound theoretical foundation, strong experimental results, and relevance to the cell biology domain. We address their comments below:
>
> ---
>
> > **Metric validity**: *Are FID/KID meaningful for cell images?*
>
> We agree that FID/KID were originally designed for natural images, but they remain **standard and valid metrics** in cell morphology prediction. They are used by all six existing baselines (e.g., IMPA, PhenDiff; see Table 5 in paper's Appendix). Our qualitative observations during development also confirmed that FID/KID effectively reflects image quality and guided model improvement.
>
> Furthermore, to ensure biological relevance, we **supplement FID/KID with mode-of-action (MoA) prediction**, which directly evaluates whether generated images preserve meaningful biological signals. As suggested by Reviewer Yg4R, we now include **MoA Accuracy, MoA Macro-F1, and MoA Weighted-F1** across in-distribution (Table 2a in paper), out-of-distribution (Table 2b  in paper), and batch effect correction settings (Table 2d in paper). All these metrics demonstrate that CellFlow consistently and significantly outperforms all baselines.
>
> |Table 2a (in-distribution evaluation)|FID/KID|MoA Acc|MoA Macro-F1|MoA Weighted-F1|
> |---|---|---|---|---|
> |Groundtruth Image|Reported in paper|72.4|69.7|72.1|
> |PhenDiff|…|52.6|33.6|52.1|
> |IMPA|…|63.7|40.2|64.8|
> |**CellFlow**|**…**|**71.2**|**49.0**|**70.7**|
>
> |Table 2b (out-of-distribution evaluation)|FID/KID|MoA Acc|MoA Macro-F1|MoA Weighted-F1|
> |---|---|---|---|---|
> |Groundtruth Image|Reported in paper|88.0|85.0|88.0|
> |PhenDiff|…|9.6|9.3|7.4|
> |IMPA|…|16.0|10.0|13.1|
> |**CellFlow**|**…**|**43.2**|**36.6**|**42.8**|
>
> |Table 2d (batch effect correction)|FID/KID|MoA Acc|MoA Macro-F1|MoA Weighted-F1|
> |---|---|---|---|---|
> |CellFlow w/ Other Batch Init|Reported in paper|48.2|32.9|48.4|
> |**CellFlow**|**…**|**71.2**|**49.0**|**70.7**|
>
> This improvement stems from our **core contribution**—**new problem formulation and solution method.** We propose modeling this task as a **distribution-to-distribution generation problem**, rather than the standard noise-to-distribution or single-to-single image prediction. This is a **conceptual shift** that aligns better with how perturbations affect heterogeneous cell populations, and allows for novel capabilities like batch effect correction and perturbation interpolation. Flow matching provides a principled and efficient tool to solve this reframed problem. While biological impact is not directly tested in this paper, CellFlow’s **strong performance and new capabilities** open the door to applications in **drug response prediction**, drug discovery, and personalized medicine in future biological studies.
>
> ---
>
> > **Interpolation plausibility**: *Do interpolations correspond to biologically meaningful transitions?*
>
> Thank you for your question. We emphasize that interpolation is a **novel and unique capability** of CellFlow, not supported by existing computational tools. Verifying intermediate cell state transition is inherently difficult, as current biotechnologies do not capture large-scale video-like morphological changes. Despite this limitation, domain experts have highlighted the potential verification of this feature—for example, in modeling dose-response curves (e.g., predicting medium-dose effects by interpolating high/low doses) or time-course dynamics (e.g., estimating 36h outcomes by interpolating 24h and 48h measurements). While our paper focuses on the **methodological foundation**, exploring these applications through biological validation is a promising direction for future work.
>
> ---
>
> > **Biological metrics**: *Could features like nucleus size be evaluated?*
>
> Thank you for the suggestion. We extracted **CellProfiler features related to nuclear size** under three perturbations known to enlarge nuclei (taxol, vincristine, and demecolcine) using the BBBC021 dataset. As shown in the table below (mean and 95% confidence interval reported), CellFlow most closely matches the real perturbed morphology in terms of nuclear size.
>
> ||**taxol**|**vincristine**|**demecolcine**|
> |---|---|---|---|
> |**Control** |1612.0 ± 39.5|1612.0 ± 39.5|1612.0 ± 39.5|
> |**Target**|2296.7 ± 190.1|2365.5 ± 125.5|2311.0 ± 136.1|
> |**PhenDiff**|1755.9 ± 138.2|1947.8 ± 70.8|2118.8 ± 102.7|
> |**IMPA**|2088.3 ± 190.8|2116.9 ± 107.1|2386.5 ± 123.9|
> |**CellFlow**|**2141.0 ± 166.6**|**2276.4 ± 115.6**|**2323.8 ± 121.9**|
>
> ---
>
> > **Reference addition**: *Please consider citing “Guided Flows for Generative Modeling and Decision Making”, which introduces classifier-free guidance in flow matching*
>
> Thank you for the suggestion. We will include this reference in the final version.
>
> ---
>
> Thank you again for your detailed feedback! We will include all of them in the revised paper.

---

> > ### Comment · Reviewer_L3Zf · 2025-04-02
> >
> > I thank the authors for their clear rebuttal. Please find my response below:
> >
> > **Metric validity:** Your argument makes sense, especially given that the metrics correlate well with MoA-based metrics as well as the biological metric (nucleus size) provided in the rebuttal.
> >
> > **Interpolation plausibility:** I am not convinced about the biological meaningfulness of the generated interpolations. By construction, I would expect that a flow-matching based method would roughly learn the shortest path in pixel space, and not necessarily a biologically meaningful path. This seems also to be the case in Figure 4, where pixels further from the nucleus are slowly dimmed, instead of e.g. the cell shrinking by retracting the membrane (but keeping well-lit pixels within the membrane). Now, I am not a cell biologist, so I cannot provide expert comments on this particular example and might be wrong for this case. Still, it seems that the claim that the method has the potential to generate interpolations that correspond to dynamic, biologically meaningful perturbation responses is unlikely in general, or at least not supported by evidence in the paper.
> >
> > **Biological metrics:** Thank you for providing these results, they are convincing and in line with the expectation from the qualitative results in the paper.
> >
> > **Concluding remarks of response:** On the one hand, the paper is a well-executed application paper, focusing on the task of predicting cell morphology changes induced by perturbations. Further, it effectively leverages the capabilities of flow matching to enable new use-cases (controlling for batch effects). On the other hand, I am not convinced about the claim that the generated interpolations can represent morphological change trajectories that are biologically meaningful, and reviewer Z8BU has expressed a similar concern in their review. Can the authors at least include (in their paper and final response) a clear overview of the limitations of this approach, and state in which conditions these interpolations follow a biologically meaningful manifold instead of "simply" interpolating in pixel-space?

---

> > > ### Author Response · Authors · 2025-04-05
> > >
> > > Dear Reviewer L3Zf,
> > >
> > > We’re glad our updated results have addressed your concerns. Your suggestions have improved the quality of our work, and we will ensure all of them are included in the revised paper.
> > >
> > > Below, we provide a detailed response to the concern about **biological plausibility of interpolated cell states.**
> > >
> > > ---
> > >
> > > **1. Limitations of Interpolation**
> > >
> > > We **fully acknowledge** that verifying whether interpolated cellular morphologies reflect biologically meaningful transitions—rather than simple pixel-space blending—is an open question. **In the submitted paper, we have already softened the language, describing interpolation as a potential capability rather than a validated outcome:**
> > >
> > > > Abstract: …CellFlow enables continuous interpolation between cellular states, providing **a potential tool** for studying perturbation dynamics…
> > >
> > > **In the final version, we will further add this sentence explicitly to the limitations section:**
> > >
> > > > Limitations: …While our method enables interpolation between cell states, we acknowledge that the biological validity of these interpolations remains unverified; establishing their plausibility will require future work involving ground-truth data and experimental validation…
> > >
> > > ---
> > >
> > > **2. Our Core Contribution**
> > >
> > > That said, we would like to re-emphasize the core contribution of this work. **Our key contribution is not the interpolation itself, but a new problem formulation: modeling cell morphology changes as distribution-to-distribution generation**. This better captures population-level effects of perturbations, and flow matching offers a principled solution for this setting. **Interpolation is an emergent capability, not the central claim of the paper.**
> > >
> > > ---
> > >
> > > **3. Opportunities for Biological Validation**
> > >
> > > We agree with the reviewer that validating the biological relevance of interpolated states is a key next step. While **existing datasets lack ground truth for intermediate states**, future work could explore:
> > >
> > > - **Dose interpolation**: Some datasets include images under multiple dosages. We can test whether **an interpolation from control to high dose passes through realistic medium-dose morphologies**.
> > > - **Timepoint interpolation**: For datasets with multiple timepoints (e.g., day 0, day 5, day 10), we can evaluate whether **interpolated images from control to day 10 recover morphology consistent with day 5**.
> > > - **Drug perturbation interpolation**: Current datasets rarely include fine-grained trajectories post-treatment. **Validating such interpolations could collect new wet-lab data, such as live-cell imaging.**
> > >
> > > ---
> > >
> > > **4. Toward More Plausible Interpolations**
> > >
> > > We appreciate your point that flow matching may yield “shortest path in pixel space” trajectories. To address this in future work, we plan to explore:
> > >
> > > - **Interpolating in latent space**: We can interpolate in latent space (e.g., via an autoencoder) rather than pixel space. This may help ensure trajectories follow a more structured biological manifold.
> > > - **Adding supervision from intermediate states**: In datasets with known intermediate points (e.g., medium dose), we can train the model to explicitly pass through those states during interpolation.
> > > - **Adding constraints**: Interpolation could be guided with additional constraints. For example, adding a GAN loss can encourage interpolated images to look like real cell images.
> > >
> > > ---
> > >
> > > **5. Context from Related Work**
> > >
> > > Latent interpolation is common in generative modeling and has been explored in biological settings using PCA or VAE-based models [1,2]. However, **few of these have been experimentally validated**, largely due to the lack of dynamic cell imaging data. Verifying such interpolations requires costly wet-lab experiments and the community’s joint effort to develop appropriate datasets and protocols.
> > >
> > > [1] Integrated intracellular organization and its variations in human iPS cells (Nature 2023)
> > >
> > > [2] Orientation-invariant autoencoders learn robust representations for shape profiling of cells and organelles (Nature Comm 2024)
> > >
> > > ---
> > >
> > > **6. Summary**
> > >
> > > In summary, we agree that the biological plausibility of interpolated trajectories is an important and open question—but one that is beyond the scope of this method-centric paper.
> > >
> > > **We hope the reviewer can recognize that CellFlow makes a substantive methodological contribution through its novel problem formulation and strong empirical performance. The interpolation capability, while not the focus, emerges naturally from our framework and is, to our knowledge, the first demonstration of such a capability in cell perturbation modeling. Biological validation of interpolations is better suited for a follow-up, biology-focused paper involving wet-lab experiments.**
> > >
> > > We will add a dedicated discussion in the appendix outlining future directions for biological validation.
> > >
> > > ---
> > > ---
> > >
> > > **Update**
> > >
> > > Thank you again for your time and constructive feedback, as well as for improving the evaluation of our work!

---

### Official Review · Reviewer_Yg4R · 2025-02-26

**Overall Recommendation:** 3

**Summary:**

This work uses a flow-based conditional generative model applied to cellular imaging, with the goal of synthetically generate, given a reference control cell and a perturbation (either chemical, genetic, or both), a novel cell image illustrating the effects of the perturbation. Cellular morphology prediction is cast as a flow problem, transforming an initial data distribution in to a final data distribution. While this task is achieved with off-the-shelf conditional flow matching models, the authors bring up a fundamental problem, related to the destructive nature of cell imaging: training data from the initial and final distributions are *not* coupled. Lack of coupling is side-stepped by observing that sampling initial cells from the same batch is sufficient to ensure proper training. Moreover, the authors note that the proposed approach allows distinguishing true biological responses to perturbation from batch-specific artifacts.

Results revolve around experiments on three datasets related to various perturbations (chemical, genetic, and both), and the proposed method is compared against two alternatives from the literature. Performance metrics are 1) image quality, 2) classification accuracy based on the mode-of-action multi-class classification problem. The proposed method improves consistently over the state of the art on both metrics. Additional ablation studies confirm the design choices made by the authors.

## POST REBUTTAL
Thanks for your detailed answers to my questions, and for the new results, which in my opinion strengthen your work.

I have modified my score accordingly.

**Claims And Evidence:**

* Claim 1: CellFlow generates high-fidelity images. This claim is partially supported by the experiments. The use of custom-made FID and KID scores is appropriate, but I am afraid that the number of generated samples (5k) is too small. Indeed, the FID score is known to have high variance when calculated on smaller sample sizes. Using only 5k images might increase the likelihood of statistical noise affecting the scores. I am afraid that the differences between methods might diminish with larger sample sizes, suggesting that the superiority of CellFlow over baselines might be less pronounced if evaluated on 50k images.

* Claim 2: The generated images capture meaningful biological patterns. This claim is partially supported by the experiments. In practical terms, an image classifier (e.g.a CNN) is trained on real, experimentally observed images of perturbed cells, where the ground-truth labels are the known MoAs of the drugs applied, basically consisting in roughly 30 classes. The MoA classification accuracy is the percentage of synthetically generated images (by the CellFlow model) for which the classifier correctly predicts the MoA, compared to the known ground-truth MoA for that perturbation.
However, class distribution is not properly discussed in the paper, and the problem is that it can be highly unbalanced, with certain classes being over-represented. Given this imbalance, accuracy as a metric could be misleading because it might be biased towards the majority classes. Alternative metrics such as macro-averaged and weighted F1-scores could be more appropriate.

* Claim 3: CellFlow generalizes to out-of-distribution perturbations never seen during training. This claim is partially supported by Table 2.b. However, the table only report image quality metrics and not the MoA metric, giving only a partial assessment of the generalization capabilities of CellFlow.

* Claim 4: CellFlow corrects batch effects by conditioning on control cells from different batches. By comparing control images with generated images, it can disentangle true perturbation-induced morphological changes from experimental batch
artifacts. This claim is marginally supported in the paper

* Claim 5: CellFlow enables bidirectional interpolation between cellular states due to the continuous and reversible nature of the velocity field in flow matching. This claim is true by construction. However, its implications are not discussed in the paper.

**Essential References Not Discussed:**

N/A

**Experimental Designs Or Analyses:**

Yes, I checked and my main comment is as follows. Section 2.2 goes great lengths in discussing the coupling problem that exists in the data. The proposed solution (in section 2.3) is valid, and simply translates into building appropriate training sets where initial and final distributions come from the same batch. My biggest concern revolves around the concept of time.

In high-content imaging experiments, cells are fixed (chemically preserved) at a specific time point after the perturbation is applied. This process halts all biological activity, effectively “freezing” the cellular state. Typically, in a well-designed experiment, both control and perturbed cells are fixed at the same time to ensure that temporal evolution does not differentially affect the two groups. However, cells are dynamic living systems, and their morphology naturally changes over time due to growth, division, apoptosis, and other metabolic activities, even in control conditions. Even when control and perturbed cells are fixed at the same time, there can be issues: cells can be at different stages of the cell cycle or physiological state, perturbed cells might respond to the treatment (chemical or genetic) at different rates depending on their initial state, leading to asynchronous responses, stochasticity in biological processes cause variability in cell morphology over time.

These aspects are marginally discussed in the paper: they are fundamental, key challenges, and I wonder what is the authors position on these issues.

**Methods And Evaluation Criteria:**

* Datasets: in my opinion the proposed datasets used in the experimental protocol are appropriate.

* Alternative methods: authors focus only on two baselines that also take control images into account. It would have been interesting to check the performance of other methods that do not do that, which would have strengthened the results and claims.

* Metrics: while I think image quality metrics to be appropriate (modulo the fact that using 5k samples might not be sufficient), MoA accuracy might not, and F1-score metrics would be preferable.

**Other Comments Or Suggestions:**

Please, check the literature! There is another work called CellFlow, published in the MLGenX workshop associated to ICLR 2024.

@inproceedings{
palma2024cellflow,
title={cellFlow: a generative flow-based model for single-cell count data},
author={Alessandro Palma and Till Richter and Hanyi Zhang and Andrea Dittadi and Fabian J Theis},
booktitle={ICLR 2024 Workshop on Machine Learning for Genomics Explorations},
year={2024},
url={https://openreview.net/forum?id=xaLXV2j8vl}
}

**Other Strengths And Weaknesses:**

This is a beautiful paper, very clearly written, and majestically illustrated!!

**Questions For Authors:**

* Would it be possible for you to report, in Table 2.a, F1-score instead of accuracy? See comments above

* Would it be possible for you to report, in Tables 1 and 2, FID and KID scores computed on more than 5k samples? Maybe 50k is too much of a computational burden, but ramping up to 20k could be feasible.

* Would you mind providing your views on the considerations above on the role of "time"?

**Relation To Broader Scientific Literature:**

This paper weights more for the biology/cell community than the machine learning community: indeed, the methodological part of the paper is based on a well-established method appeared several years ago, and there are no major contributions apart from the discussion in section 2.2 and section 2.3.

From the biology/cell literature point of view, I think this paper might represent an interesting advancement, but the experimental protocol and results are somehow shallow in this respect: focusing only on MoA accuracy, while important (and possibly improvable), does not give a sense of the real impact of this work in a, e.g., drug design context (as per the introduction in the paper).

**Theoretical Claims:**

The proof for Proposition 1 appears correct to me.

---

> ### Author Rebuttal · Authors · 2025-04-01
>
> We thank Reviewer Yg4R for their appreciation of the paper’s beautiful writing and illustration, proper experimental design and evaluation, and strong relevance to the biology and cell imaging community. We address their comments below:
>
> ---
>
> > **Claim 1 (effect of sample size on FID/KID):** *Could FID/KID score differences diminish with larger sample sizes?*
>
> We added evaluations of different sample sizes on BBBC021 (1K–5K, only 6K test images available) and JUMP (10K–20K). As shown, **CellFlow consistently outperforms all baselines across sample sizes**, with 30–45% relative improvement, demonstrating robustness.
>
> ||1K FID|2.5K FID|5K FID|10K FID|20K FID|1K KID|2.5K KID|5K KID|10K KID|20K KID|
> |-|-|-|-|-|-|-|-|-|-|-|
> |PhenDiff|71.3|64.3|49.5|47.5|46.1|2.55|3.68|3.10|4.95|5.09|
> |IMPA|52.4|41.4|33.7|14.0|12.9|3.20|3.38|2.60|1.04|1.05|
> |CellFlow|34.7|25.2|18.7|8.5|7.5|1.67|1.90|1.62|0.63|0.63|
>
> ---
>
> > **Claim 2 (class imbalance in MoA prediction)**: *Could you report macro and weighted F1 scores in Table 2.a?*
>
> We now report **Macro-F1 and Weighted-F1** scores for MoA classification. CellFlow outperforms all baselines not only in accuracy but also in F1 metrics, addressing class imbalance concerns.
>
> ||Acc|Macro-F1|Weighted-F1|
> |-|-|-|-|
> |Groundtruth Image|72.4|69.7|72.1|
> |PhenDiff|52.6|33.6|52.1|
> |IMPA|63.7|40.2|64.8|
> |CellFlow|71.2|49.0|70.7|
>
> ---
>
> > **Claim 3 (OOD generalization)**: *Why are MoA metrics not reported for OOD perturbations in Table 2.b?*
>
> We now **include MoA metrics** for OOD perturbations. CellFlow significantly outperforms baselines in this setting as well, reinforcing the model’s generalization capability.
>
> ||Acc|Macro-F1|Weighted-F1|
> |-|-|-|-|
> |Groundtruth Image|88.0|85.0|88.0|
> |PhenDiff|9.6|9.3|7.4|
> |IMPA|16.0|10.0|13.1|
> |CellFlow|43.2|36.6|42.8|
>
> ---
>
> > **Claim 4 (batch effect)**: *Claim on batch correction is marginally supported.*
>
> CellFlow explicitly conditions on control images from the **same batch** to correct batch effect. In the table below (an extension of Table 2.d in the paper, with MoA evaluation added), we compare generation quality and MoA classification performance when using **same-batch** versus **other-batch** control initialization. Using controls from the same batch yields significantly better results, highlighting their importance for correcting batch effects and capturing true perturbation signals.
>
> ||FID$_o$|FID$_c$|KID$_o$|KID$_c$|MoA Acc|MoA Macro-F1|MoA Weighted-F1|
> |-|-|-|-|-|-|-|-|
> |CellFlow w/ Other Batch Init|23.7|71.9|2.08|2.09|48.2|32.9|48.4|
> |CellFlow w/ Same Batch Init|18.7|56.8|1.62|1.59|71.2|49.0|70.7|
>
> ---
>
> > **Methods and evaluation (alternative methods)**: *Comparison with more baselines would strengthen the claim.*
>
> Cell morphology prediction is a new task with only six baselines (Table 5 in Appendix). We included the **only two published methods** using control images; others are unpublished, lack code, or omit controls. We also added MorphoDiff (ICLR 2025), a diffusion-based method. Under our setup, CellFlow outperforms it in image quality and MoA metrics.
>
> ||FID$_o$|KID$_o$|FID$_c$|KID$_c$|MoA Acc|MoA Macro-F1|MoA Weighted-F1|
> |-|-|-|-|-|-|-|-|
> |MorphoDiff|65.8|7.99|114.1|7.97|38.3|24.5|34.2|
> |CellFlow|18.7|1.62|56.8|1.59|71.2|49.0|70.7|
>
> ---
>
> > **Experimental designs (time effect)**: *How does the method handle issues like fixation and cell cycle variation?*
>
> We appreciate this insightful question. Cell morphology naturally varies over time due to asynchronous cell cycles and stochastic responses. CellFlow models **distribution-level transformations** rather than single-cell alignments, enabling it to **average out temporal and biological variability and capture population-level perturbation effects**. We believe this distribution-to-distribution approach offers a principled solution to handling temporal variation in image-based profiling.
>
> ---
>
> > **Claim 5 (interpolation use case)**: *Implications of bidirectional interpolation not discussed.*
>
> > **Relation to broader science (real-world impact)**: *What is the impact of this work for drug design or discovery?*
>
> We appreciate the reviewer’s comment. Our core contribution is a **new problem formulation**—modeling perturbation effects as a distribution-to-distribution transformation—and a **principled solution** via flow matching. This leads to significantly stronger performance and unlocks several novel capabilities relevant to biology: for example, batch effect correction helps isolate true biological signals, and perturbation interpolation enables exploration of intermediate drug doses or timepoints. While full biological validation is beyond the scope of this paper, these capabilities lay the groundwork for future impact in drug design & discovery.
>
> ---
>
> > **Other comments (naming)**
>
> We will **rename our method to CellFlux** to avoid confusion. Thank you for bringing this to our attention.
>
> ---
>
> Thank you again for your detailed feedback! We will include all of them in the revised paper.

---

> > ### Comment · Reviewer_Yg4R · 2025-04-02
> >
> > Dear authors, thank you for the detailed responses to my questions and observations. Overall, the new results better support the claims in the paper, and for this reason I will raise my score.
> >
> > Good job!

---

> > > ### Author Response · Authors · 2025-04-05
> > >
> > > Dear Reviewer Yg4R,
> > >
> > > We are glad that our updated results and clarifications have addressed your concerns. Your suggestions have improved the quality of our work, and we will ensure that all of them are included in the revised paper. Thank you again for your time and constructive feedback!

---

### Official Review · Reviewer_Z8BU · 2025-03-07

**Overall Recommendation:** 4

**Summary:**

This work introduces CellFlow, an image-generative model designed to simulate cellular morphology changes induced by chemical and genetic perturbations. It leverages flow matching, a generative modeling technique, to learn a distribution-to-distribution map that transforms unperturbed cell states into perturbed ones. CellFlow is evaluated on chemical, genetic, and combined perturbation datasets, showing significant improvements in FID scores and mode-of-action prediction accuracy compared to existing methods. Due to the nature of flow matching, the model allows for continuous interpolation between cellular states, offering a tool for studying perturbation dynamics and also the recovery dynamics if the transformation is considered in reverse time.

## update after rebuttal
Thank you for the additional experiments and clarifications in response to my feedback. The updated results addressed many of my concerns, and I appreciate the authors' efforts to improve the quality of the work based on my suggestions. As a result, I have revised my evaluation from a 3 (Weak Accept) to a 4 (Accept), reflecting my updated assessment of the paper's strengths and improvements. I also commend the authors for their commitment to incorporating all suggestions into the revised version, which will likely enhance the overall contribution of the work.

**Claims And Evidence:**

The usefulness of the interpolated dynamics from a biological perspective might be limited as it could be the mathematical artifacts of the Flow Matching formulation rather than a reflection of a concrete biological process.

**Essential References Not Discussed:**

-

**Experimental Designs Or Analyses:**

The considered datasets serve the purpose of this paper as they cover different types of perturbations and various cells.

**Methods And Evaluation Criteria:**

The evaluation makes sense for conditional image generation in cell painting.

**Other Comments Or Suggestions:**

-

**Other Strengths And Weaknesses:**

## Strengths
- Innovative Approach: Although flow matching is relatively explored as a conditional distribution matching framework, its application in cellular morphology seems novel.
- Performance: The model demonstrates significant improvement (35% in FID scores and 12% in mode-of-action prediction accuracy) over existing methods.
- Biological Relevance: The generated images are biologically meaningful and capture perturbation-specific morphological changes.
Potential Applications: The ability to interpolate between cellular states could be a valuable tool for studying perturbation dynamics even though the dynamics may not be biologically plausible.
## Weaknesses:
- Technical Novelty: Although the application of Flow Matching in this context may be novel, the technical novelty of the work is limited to using an existing method in a new domain. In the absence of a major technical novelty, a more extensive experimental setup is expected to show the utility of the method as an application-driven work.
- Complexity: The computational complexity of the method is not discussed. This could bring more clarity on the scalability of the approach.
- Control Wells: The importance of control wells is highlighted, but the paper does not detail how variations in control conditions are managed. For example, the wells without perturbations may exhibit different batch effects simply because of being in different wells.

**Questions For Authors:**

- How does CellFlow handle variations in control conditions across different batches, and how does this impact the model's predictions?
- What are the computational requirements for training CellFlow, and how does it scale with larger datasets or more complex perturbations?
- The nature of the method that takes perturbation as input to the velocity network allows for interpolation over the perturbation axis. Given the complexity of biological systems, can such interpolation be trusted? Can authors provide the assumptions about the underlying system that justify trusting the predicted effect of unseen perturbations?
- It is imaginable that there is some transfer learning of the effect of perturbations from one dataset to another. For example, when a perturbation is not present in a dataset, observing its effect in another dataset could be helpful to fill the gap in the target dataset. Does CellFlow allow for such transfer of knowledge across datasets?

**Relation To Broader Scientific Literature:**

- The use of flow matching in a novel context.
- Addressing an important problem in biology and drug discovery which is the effect of perturbations on cells.

**Theoretical Claims:**

There is no major theoretical claim in this work. Proposition 1 makes intuitive sense and its proof in the appendix also seems correct.

---

> ### Author Rebuttal · Authors · 2025-04-01
>
> We thank Reviewer Z8BU for recognizing the paper’s proper evaluation criteria, sound theoretical claims, well-designed experiments, comprehensive supplementary material, as well as its innovative approach, strong performance, and biological relevance. We address their concerns below:
>
> ---
>
> > **Technical novelty**: *How does the work contribute technically beyond applying flow matching to a new domain?*
>
> We thank the reviewer for this opportunity to clarify. Our key contribution lies not in modifying flow matching itself, but in **reformulating the problem of cell morphology prediction**: we propose modeling this task as a **distribution-to-distribution generation problem**, rather than the standard noise-to-distribution or single-to-single image prediction. This is a **conceptual shift** that aligns better with how perturbations affect heterogeneous cell populations, and allows for novel capabilities like batch effect correction and perturbation interpolation. Flow matching provides a principled and efficient tool to solve this reframed problem.
>
> ---
>
> > **More experimental settings**: *In the absence of major technical novelty, a more extensive experimental setup is expected.*
>
> We would like to highlight that our experimental setup is **exhaustive** compared to existing works:
>
> - **Perturbations**: chemical, genetic, and combined.
> - **Datasets**: BBBC021, RxRx1, and JUMP.
> - **Settings**: both **in-distribution (ID)** and **out-of-distribution (OOD)** generalization.
> - **Evaluation**: includes **overall FID / KID**, **conditional FID / KID**, and **MoA classification**.
> - **Capabilities**: our model supports **batch effect correction** and **perturbation interpolation**—capabilities that existing methods do not support.
>
> ---
>
> > **Computational complexity**: *What are the training requirements and scalability for larger datasets or complex perturbations?*
>
> CellFlow is **computationally efficient** and scales **linearly** with dataset size. We already provided the training details in *Appendix C*: “Models are trained for 100 epochs on 4 A100 GPUs ... requiring 8, 16, and 36 hours for BBBC021, RxRx1, and JUMP, respectively”.
>
> ---
>
> > **Control wells**: *How does CellFlow handle variations in control conditions across batches, and how does this impact the model's predictions?*
>
> CellFlow explicitly conditions on control images from the same batch to account for batch-specific variations *(discussed in Section 2.3 in the paper)*. This means that for each perturbed sample, the model is provided with the corresponding unperturbed (control) cells from the **same experimental batch**, enabling it to distinguish true perturbation effects from batch-induced variability.
>
> In the table below *(an extension of Table 2d in the paper, with MoA evaluation added as suggested by Reviewer Yg4R)*, we compare generation quality and MoA classification performance when using **same-batch** versus **cross-batch** control initialization. Using controls from the same batch yields significantly better results, highlighting their importance for correcting batch effects and capturing true perturbation signals.
>
> ||FID$_o$|FID$_c$|KID$_o$|KID$_c$|MoA Accuracy|MoA Macro-F1|MoA Weighted-F1|
> |-|-|-|-|-|-|-|-|
> |Condition on control wells from different batch|23.7|71.9|2.08|2.09|48.2|32.9|48.4|
> |Condition on control wells from same batch|18.7|56.8|1.62|1.59|71.2|49.0|70.7|
> |Relative improvement|+21.1%|+21.0%|+22.1%|+23.9%|+47.7%|+49.0%|+46.1%|
>
> ---
>
> > **Interpolation trustworthiness**: *Can interpolation in perturbation space be trusted biologically?*
>
> We emphasize that interpolation is a **novel and unique capability** of CellFlow, not supported by existing computational tools. Verifying intermediate cell states is inherently difficult, as current biotechnologies do not capture large-scale video-like morphological changes. Despite this limitation, domain experts have highlighted the potential verification of this feature—for example, in modeling dose-response curves (e.g., predicting medium-dose effects from high/low doses) or time-course dynamics (e.g., estimating 36h outcomes from 24h and 48h measurements). While our paper focuses on the **methodological foundation**, exploring these applications through biological validation is a promising direction for future work.
>
> ---
>
> > **Cross-dataset transfer**: *Can CellFlow generalize perturbation effects across datasets?*
>
> Thank you for the insightful question. We conducted a **transfer experiment** by applying a CellFlow model trained on BBBC021 to RxRx1 and JUMP images. These datasets lack ground-truth perturbed counterparts, making quantitative evaluation infeasible. However, we observed that CellFlow is able to transfer and apply perturbation effects despite substantial domain shifts, with **qualitative examples** available at https://anonymous.4open.science/r/CellFlow-Rebuttal/cross_dataset.png.
>
> ---
>
> Thank you again for your detailed feedback! We will include all of them in the revised paper.

---

> > ### Comment · Reviewer_Z8BU · 2025-04-07
> >
> > Thank you for the response and additional experiments. The clarifications addressed some of my concerns and I have updated my score accordingly.

---

> > > ### Author Response · Authors · 2025-04-07
> > >
> > > Dear Reviewer Z8BU,
> > >
> > > We’re glad that our updated results have addressed your concerns and that you are willing to improve your evaluation of our work as a result. Your suggestions have enhanced the quality of our work, and we will ensure that all of them are included in the revised paper. Thank you again for your time and constructive feedback!

---

### Decision · Program_Chairs · 2025-05-01

**Decision:**

Accept (poster)

**Comment:**

The paper utilizes state of the art generative modelling method for in-silico biological experiments that may have an important impact for future research in this direction. The reviewers are in agreement that even though the work has some limitations, it is overall well executed and is a valuable contribution to the field.